# TEN-DM: TOPOLOGY-ENHANCED DIFFUSION MODEL FOR SPATIO-TEMPORAL EVENT PREDICTION

**Yuxin Liu** [1]   **Kaiming Wang** [1]   **Chenguang Yang** [1]   **Yulia R. Gel**[2]   **Yuzhou Chen**[1]
[1]Department of Statistics, University of California, Riverside
[2]Department of Statistics, Virginia Tech
{yliu1020, kwang280, cyang314, yuzhou.chen}@ucr.edu
ygl@vt.edu

## ABSTRACT

Spatio-temporal point process (STPP) data appear in many domains. A natural way to model them is to describe how the instantaneous event rate varies over space and time given the observed history which enables interpretation, interaction detection, and forecasting. Traditional parametric kernel-based models, while historically dominant, struggle to capture complex nonlinear patterns. In contrast, deep learning methods leverage the representational power of neural networks to aggregate historical events and integrate spatio-temporal point processes. However, existing deep learning methods often process space and time independently, overlooking the spatio-temporal dependencies. To address this limitation, we propose a novel method called Topology-ENhanced Diffusion Model (TEN-DM), including two key components namely spatio-temporal graph construction and multimodal topological feature representation learning. Further, we use temporal query technique to effectively capture periodic temporal patterns for learning effective temporal representations. Extensive experiments show the effectiveness of TEN-DM on multiple STPP datasets compared to state-of-the-art methods.

## 1 INTRODUCTION

Stochastic point processes are loosely speaking random sets of points (marks) scattered over some domain. Such processes appear in a wide range of natural and manmade phenomena and can be also used to characterize various human activities, with applications ranging from earthquake occurrence to emergency calls to heart beat. Some point processes can be put in correspondence with an index. If such index represents time, we call this stochastic process a temporal point process (TPP) like, for instance, high-frequency trading order book events in finance or patient's decline towards septic shock in medicine which are often modeled by Hawkes processes (Lima, 2023; Kuang et al., 2024; Laub et al., 2025). If the points live in some $d$-dimensional domain (e.g., Euclidean space or a manifold), for example, patterns of Ice Age archaeological sites Jayalath et al. (2015), spatial arrangements of trees and animals in ecology (Samarasekara et al., 2025) or distributions of young stellar objects in astronomy (Retter et al., 2019), we call it a spatial point process (SPP). In turn, spatio-temporal point processes (STPP) extend these concepts by considering stochastic processes that integrate both spatial and temporal dimensions. STPP allow us to consider phenomena in which events occur within a spatial domain, with the time of occurrence serving as a distinguishing feature (mark) associated with each event, with applications ranging from earthquake tracking to crime detection to monitoring infectious diseases (Zhu & Xie, 2022; Bernabeu et al., 2025), just to name a few. SPP, TPP and STPP have been a longstanding area of research within statistical sciences (Cox & Isham, 1980; Daley & Vere-Jones, 2003; Diggle, 2013). However, such more traditional statistical approaches predominantly either impose some parametric model restrictions or tend to rely heavily on unrealistic assumptions about the event sequences' generative processes, exhibiting limited abilities to scale for long historical records and massive event sets.

In turn, numerous recent efforts have been dedicated to developing DL for modeling TPPs and STPPs (see the most recent comprehensive overview by Cheng et al. (2025)). Some of the earlier thrusts in this direction include combination of an event encoder, aggregation encoder, and decoder parametrization for event sequence prediction (Du et al., 2016; Shchur et al., 2021). The more

recent techniques advance the concept of deep STPPs which encompass such methods as deep kernels, neural latent processes with transformers, and deep generative models (Zhu et al., 2020; Chen et al., 2021a; Zhou et al., 2022; Cheng et al., 2025). Despite these advances, due to inherent non-stationarity and complex interplay within and between time and space dimensions, existing DL tools often exhibit limited capabilities to capture intrinsic structural organization of the underlying STTPs and to distill intricate latent spatio-temporal interdependencies, especially under sparse regimes, which leads to deficiencies in predictive performance (Cheng et al., 2025). Our paper is, hence, motivated by the following tightly interwoven questions: *How can we describe the complex spatio-temporal STTP interdependencies, especially under sparse and noisy scenarios?* and *How can we distill latent structural STTP characteristics that play a particularly important role for predictive tasks?*

We argue that these fundamental questions in STTP modeling can be approached by fusing the raising paradigm of diffusion models with the emerging tools from computational topology, particularly, the concepts of zigzag persistence. **Why graphs?** As shown by a number of recent studies (Jin et al., 2024), graph abstraction offers a flexible and versatile framework to describe higher-order interdependencies in multivariate spatio-temporal processes which otherwise cannot be systematically assessed by more traditional methods. Despite this success and despite the existence of STTPs on graphs and manifolds, graph abstractions have never been used to model STTPs. We fill this gap by designing a STPP graph construction strategy with different views that convert STPP into a graph abstraction and learning node (i.e., event) embeddings. By leveraging temporal query and self-attention on data in temporal dimension, we then capture period patterns and temporal-wise dependencies. This allows us to enhance prediction in spatial and temporal domains. **Why topology and why zigzag persistence?** In a nutshell, computational topology extracts shape properties of the data that are intact under continuous transformations. Integrating such extracted topological descriptors to DL has shown to result in enhanced model performance and robustness gains. Zigzag persistence (ZP) advances these ideas toward distilling the most essential shape signatures of the data that manifest over time. While ZP has been studied in conjunction with graph diffusion, the utility of ZP for modeling STTPs has never been explored. To leverage such important time-aware shape information, we transform the observed STPP to a time series of images and then, armed with ZP, learn the most essential topological characteristics that reveal over time. Finally, **Why diffusion?** Thanks to their advanced capabilities to capture complex relational structures within the observed data, diffusion models have recently emerged as a new powerful machinery for a variety of downstream tasks, from anomaly detection to prediction. While there are a number of studies on diffusion models for TTPs (Lüdke et al., 2023; Zhang et al., 2024) to the best of our knowledge, neither of them yet consider diffusion for STTP. As such, this paper advances STTP modeling along multiple directions: by leveraging zigzag-enhanced diffusion and STTP graph representation, we propose a novel versatile Topology-ENhanced Diffusion Model (TEN-DM) for capturing complex spatio-temporal dynamics of STTP under sparse and noisy regimes.

In summary, the paper makes the following key contributions:

- We design (i) a STPP graph construction and learning (GCL) module to preprocess STPP into the graph format, enabling the GNNs to learn complex spatio-temporal interactions; (ii) a novel temporal topological learning (TTL) framework coupled with cubical zigzag persistence, which captures topology-aware spatio-temporal information over STPP; and (iii) a temporal query-guided self-attention mechanism (TQ-SA) to capture temporal dependencies.

- Based on the GCL, TTL, and TQ-SA, we introduce TEN-DM, a novel diffusion model to address the *"dynamic spatio-temporal dependencies learning dilemma"* in STPP.

- Extensive experiments on 5 real-world STPP datasets show the proposed TEN-DM achieves state-of-the-art prediction performance in both spatial and temporal dimensions.

## 2 TECHNICAL BACKGROUND

**Notations & Problem Formulation.** We are given a sequence of spatio-temporal events $X = \{x_i \mid i = 1, 2, \ldots, N\}$ whose number of events $N$. Each event is described as $x_i = (t_i, g_i)$, where $t_i$ denotes $i$-th occurrence time, $g_i$ denotes $i$-th geospatial information (e.g., latitude, longitude, or zipcode), and $0 < t_1 < \cdots < t_{\mathcal{T}} < T$ (i.e., a sequence of strictly increasing arrival times). The goal

of forecasting model $\mathcal{F}_\theta$ with weights $\theta$ to predict future spatial and temporal information based on the history until time $t$ denoted as $\mathcal{H}_t = \{\boldsymbol{x}_1, \ldots, \boldsymbol{x}_n\}_{t_n < t}$, i.e., $\hat{\boldsymbol{x}}_{t+1} = \mathcal{F}_\theta(\mathcal{H}_t)$.

**Diffusion Models.** Diffusion models are probabilistic generators that learn a data distribution by corrupting samples with Gaussian noise in a forward Markov chain and then training a neural network to iteratively denoise in reverse. This framework has exerted a significant influence on state-of-the-art results in computer vision (Rombach et al., 2022; Saharia et al., 2022; Ho et al., 2022) and natural language processing (Gong et al., 2023; He et al., 2023; Li et al., 2022). Given their ability to capture data distributions, diffusion models are increasingly studied for spatio-temporal data forecasting and generation in a variety of scenarios. DiffSTG (Wen et al., 2023) adapts diffusion to spatio-temporal graphs and introduces a UNet style temporal module with graph convolutions. KSTDiff (Zhou et al., 2023) tackles urban flow generation by combining a region-customized diffusion process guided by a learned volume estimator with a knowledge graph-enhanced denoising network. Dyffusion (Rühling Cachay et al., 2023) introduces a dynamics informed diffusion model by embedding temporal dynamics into the diffusion steps and training a stochastic time conditioned interpolator with a predictor. Diff-RNTraj (Wei et al., 2024) focus on vehicle trajectory data, which pretrains continuous embeddings of road information as denoising condition and decodes back with a spatial-validity loss. ControlTraj (Zhu et al., 2024) develops a autoencoder that learns road segment embeddings and combines road-network topology constraints, merging them into a geographic UNet to guide the denoising process. Finally, Chen & Gel (2025) propose to integrate the notion of zigzag persistence into the graph diffusion framework, with application to spatio-temporal forecasting and graph classification.

**Point Processes.** Point processes are widely used to model sequences of discrete events across diverse domains (Daley & Vere-Jones, 2008; Reiss, 2012; Karr, 2017). Classical TPP models focuses on conditional intensity function (Rasmussen, 2018), including Poisson process (Kingman, 1992), Hawkes process (Hawkes, 1971) , and self-correcting process (Isham & Westcott, 1979). The simple patterns of occurrence can be captured by the classical TPP models, while the neural TPP models can perform better in capturing complex dependencies (Shchur et al., 2021). Decoupled Marked Temporal Point Process (MTPP) (Song et al., 2024) uses Neural ODEs to decouple the influence of each past event into its own latent continuous trajectory. Neural Jump-Diffusion TPP (NJDTPP) (Zhang et al., 2024) proves the equivalence of stochastic differential equations (SDEs) for classical TPPs, and uses neural jump-diffusion SDE (NJDSDE) which provides a unified SDE view with theoretical footing. For the SPP, it is well introduced in Moller & Waagepetersen (2003); Illian et al. (2008). Continuous normalizing flows (CNF) and Time-Varying CNF (TVCNF) can be used for modeling spatial distribution where the latter considers the dependence on the timestamps (Chen et al., 2018; 2021a). Beyond TPP and SPP, STPP takes spatial and temporal information into consideration. Classical STPPs are extended from the point process including (in)homogeneous Poisson process (Daley & Vere-Jones, 2003), Neyman–Scott process (Gabriel & Diggle, 2009), inhibition process (Gabriel et al., 2013), strauss process (Cronie & Van Lieshout, 2015), and Cox process (Cox, 1955; Diggle, 2013; Diggle et al., 2013). Recent neural approaches extend STPPs along two main directions, i.e., Influence-kernel–based models and Intensity-based models (Cheng et al., 2025). Deep Non-stationary Kernel (DNSK) (Dong et al., 2020) develops a novel and general low-rank decomposition to approximate the influence kernel and representation through deep neural networks. However, real-world STPP often exhibit complex and nonstationary spatio-temporal dependencies, which leads to significant challenges in accurately predicting spatio-temporal events. In contrast, our TEN-DM introduces spatio-temporal geometric and topological learning paradigm, which can effectively introduce the graph structural and dynamic topological information into the diffusion model, thereby being able to capture complex spatio-temporal dependence between discrete events.

## 3  METHODOLOGY

To address the limitations of existing STPP approaches and leverage the strengths of geometric and topological representation learning, we propose TEN-DM, a unified diffusion model framework that integrates structural, temporal, and topological information for enhanced spatial and temporal forecasting. As illustrated in Figure 1, the framework comprises three core components.

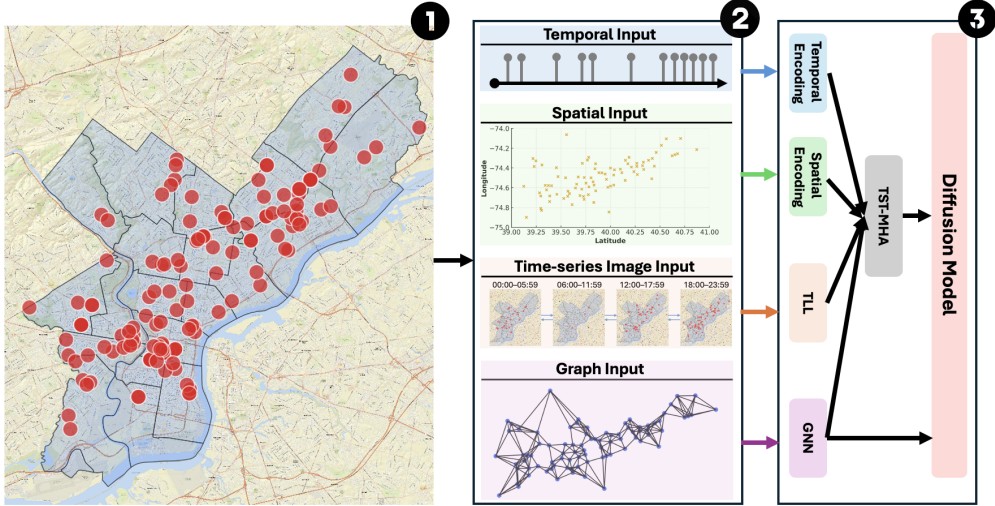

Figure 1: Overview of the TEN-DM model: (1) a sequence of events; (2) extracted temporal, spatial, time-series image, and graph information; (3) key components including temporal encoding, spatial encoding, temporal topological learning module (TLL), graph representation learning (GNN), and topology-guided spatio-temporal multi-head attention (TST-MHA).

## 3.1 SPATIO-TEMPORAL POINT PROCESS GRAPH LEARNING

Compared with existing approaches which neglect spatio-temporal point process inherent interactions between different event properties, to address this limitation, we propose a new strategy to capture the nuanced spatial and temporal relationships by generating a graph structure and learn the corresponding high-level graph representation. There exist multiple types of edges between nodes (i.e., events), and each type of edge has a different role and impact on node representation learning. For example, for the graph representation learning in crime incidents, different types of relationships between crime incidents exhibit various dependency semantics such as thefts in specific census tracts can be temporally linked to seasonal trends (e.g., spikes during holidays), robberies tied to highways or transit stations, and burglaries are often higher in neighborhoods with high poverty rates and housing instability. Therefore, in this section, we aim to answer the question: *How to build a STPP graph by fully utilizing the rich multi-semantic information?*. We first generate multiple graphs by differentiating the types of edge connections between nodes in the heterogeneous graph. Then we aggregate the relation-guided graph structural information with different importance weights. We denote our generated graph as $\mathcal{G}_r = (\boldsymbol{A}_r, \boldsymbol{X}_r)$ (where $r = \{1, \dots, \mathcal{R}\}$; note that, in our study, $\mathcal{R} = 3$ includes time, latitude, and longitude information) with $r$-th node feature matrix $\boldsymbol{X}_r$ and the adjacency matrix of $r$-th graph $\boldsymbol{A}_r$. To achieve this, we first construct a $\epsilon$-graph $\mathcal{G}_r$ (Calder & Trillos, 2022). Specifically, we quantify similarity $\boldsymbol{s}_{uv}$ between events $u$ and $v$ as follows:

$$
\begin{aligned}
\boldsymbol{s}_{uv}^r &= (\boldsymbol{x}_u^r \odot \boldsymbol{x}_v^r)/(||\boldsymbol{x}_u^r||_2||\boldsymbol{x}_v^r||_2), \\
e_{uv}^r &= \{(u,v) \mid \boldsymbol{s}_{uv}^r > \mathscr{R}^r\},
\end{aligned}
\tag{1}
$$

where $\odot$ denotes dot product. In this work, we use cosine similarity to calculate event similarity. By using a cell list to find event pairs that are within a given cut-off distance, we can efficiently solve the problem with a time complexity $\mathcal{O}(kN)$ where $k$ is the maximal number of neighbors within the radius. Hence, node interactions with various relation semantics will have different structural characteristics. To capture such multi-typed node dependencies, we assign different weights to different adjacency matrices and aggregate $\mathcal{R}$ edge-type-specific adjacency matrices as $\mathbb{A} = \sum_{r=1}^{\mathcal{R}} \alpha_r \boldsymbol{A}_r$. Note that the set of importance weights $\{\alpha_r\}$ are updated adaptively during training.

To deliver high-level graph-structured information into our diffusion model framework, we pretrain a GNN with the node features $\mathbb{X}$ (where $\mathbb{X} = \oplus(\boldsymbol{X}_1, \boldsymbol{X}_2, \dots, \boldsymbol{X}_{\mathcal{R}})$ and $\oplus$ denotes the concatenation) and joint adjacency matrix $\mathbb{A}$, and adopt a pooling layer (i.e., Pool($\cdot$)) to get the graph representation

which can be formalized as:

$$\boldsymbol{o}_{\mathcal{G}} = \text{Pool}(\text{GNN}(\mathbb{X}, \mathbb{A})). \tag{2}$$

## 3.2 TIME-SERIES IMAGE TOPOLOGICAL REPRESENTATION LEARNING

To enhance spatio-temporal prediction by incorporating dynamic topological information, we create time-series images for spatio-temporal point process data and propose an effective method, i.e., dynamic image topology learning module that captures dynamic visual scenes.

**Formulating time-series image representation by the sequence of events.** Given a sequence of events $\boldsymbol{X} = \{\boldsymbol{x}\}_{i=1}^{T}$, we first divide $\boldsymbol{X}$ into patches which can be either overlapped or non-overlapped. Here we set the patch length (i.e., scale) to be $P$ and the stride to be $S$ (i.e., the non-overlapping region between two neighboring patches). Then we can obtain a sequence of patches which is denoted by $\hat{\boldsymbol{X}} = \{\boldsymbol{x}^{(1)}, \boldsymbol{x}^{(2)}, \dots, \boldsymbol{x}^{(N)}\} \in \mathbb{R}^{P \times N}$ where $\boldsymbol{x}^{(i)} \in \mathbb{R}^{P}$ and $N = \lfloor (T - P)/S \rfloor + 2$. After that, for each patch $i$, we create an image $m_i \in \mathbb{R}^{H \times W}$ which is a 2D grid and clipped to the target data region. For instance, given the Philadelphia crime incident data, the 2D grid is bounded by the set of latitude (min: 39.86; max: 40.14) and longitude (min: −75.28; max: −74.95). Within the patch $i$, we rasterize the events' geo-coordinates onto the 2D image by recording as each pixel's value the associated temporal attribute. Specifically, we convert the spatial locations of events into a grayscale image, where each pixel corresponds to a discretized spatial cell. Pixels containing at least one event are assigned a value of 1 (white), while pixels without events are assigned 0 (black). Thus, we can create a binary image that encodes the spatial distribution of events within the patch. That is, $m_{x_j, y_j}^{(i)} = \boldsymbol{x}^{(i)}[j]$ which represents $j$-th timestamp in $i$-th patch $\boldsymbol{x}^{(i)}$. Thus, we can generate the time-series image data $\boldsymbol{M}$ as a series of images $\{m^{(1)}, m^{(2)}, \dots, m^{(N)}\}$.

**Topology learning on time-series image data.** Persistent homology (PH) is a branch in topological data analysis which tracks the evolution of the various data shape patterns along various user-selected geometric dimensions (Zomorodian & Carlsson, 2004; Edelsbrunner & Morozov, 2012). Despite the generality of simplicial complexes in PH, cubical complex is a more natural representation for 2D images or 3D volumes. See Appendix A for formal definition of the cubical complex.

**Definition 3.1** *Let $\mathscr{K}$ be a cubical complex, and suppose $f : Q_{\mathscr{K}} \mapsto \mathbb{R}$, where $Q_{\mathscr{K}}$ denotes the set of elementary cubes in $\mathscr{K}$, satisfies (i) $f(Q') \leq f(Q)$ whenever $Q'$ is a face of $Q$. Let $\mathscr{K}(\alpha) = f^{-1}((-\infty, \alpha])$, and notice that (i) implies $\mathscr{K}(\alpha)$ is a subcomplex of $\mathscr{K}$ for every $\alpha \in \mathbb{R}$. Taking $\alpha_1 < \alpha_2 < \cdots < \alpha_n$ to be the values of $f$ on the cubes of $\mathscr{K}$ and denoting $\mathscr{K}(\alpha_i) = \mathscr{K}_{\alpha_i}$, we say that the following increasing sequence of subcomplexes is a filtration associated with $f$*

$$\varnothing = \mathscr{K}_{\alpha_0} \subset \mathscr{K}_{\alpha_1} \subset \cdots \subset \mathscr{K}_{\alpha_n} = \mathscr{K}.$$

In our setup, we use the lower-star filtration to extract the topological information encoded in 2D images. However, given time-series image data, standard persistence algorithms capture only independent spatial topological information, and unaware of the temporal topological information and complex spatio-temporal dependencies. Furthermore, beyond capturing dynamic structures and temporal dependencies, it is equally important to learn multi-scale temporal information, i.e., understanding temporal information by leveraging multi-scale time-related data such as time units (e.g., minutes, hours, and days). To address the above two challenges, we propose a more flexible temporal topology learning (TTL) framework coupled with zigzag persistence which is capable of capturing vital time-aware topological information on time-series images across different time scales.

**Zigzag persistence.** Zigzag persistence (ZP) is a special type of quiver representations and generalizes conventional PH by enabling the analysis of topological spaces connected through inclusions in both forward and backward directions (Carlsson & De Silva, 2010). Unlike the above standard PH which requires a nested sequence of spaces, ZP can accommodate more flexible filtrations which makes it particularly well suited for capturing the evolving topological structure of image time-series data. This capability has led to growing interest in ZP across a range of data analysis tasks involving time-varying signals. Specifically, for a time sequence of images $\{m^{(1)}, m^{(2)}, \dots, m^{(N)}\}$, the zigzag filtration over image snapshots is constructed by the bidirectional arrows as follows:

$$m^{(1)} \hookrightarrow m^{(1)} \cup m^{(2)} \hookleftarrow m^{(2)} \hookrightarrow \cdots \hookleftarrow m^{(N-1)} \hookrightarrow m^{(N-1)} \cup m^{(N)} \hookleftarrow m^{(N)}$$

To compute the homology groups and the corresponding topological feature of the zigzag filtration, for each timestamp $i$, we first apply the cubical complex (i.e., with lower star filtration) to the image $m^{(i)}$ at time step within the $i$-th patch and construct a simplicial complex $\mathscr{K}^i$. Then we compute the union of two adjacent simplicial complexes $\mathscr{K}^{(i)}$ and $\mathscr{K}^{(i+1)}$ denoted by $\mathscr{K}^{(i,i+1)} = \mathscr{K}^{(i)} \cup \mathscr{K}^{(i+1)}$, i.e., we include a simplex $\rho \in \mathscr{K}^{(i,i+1)}$ if and only if $\rho \in \mathscr{K}^{(i)}$ or $\rho \in \mathscr{K}^{(i+1)}$ which preserves features that appear (born) or disappear (die) across time steps. Leveraging the zigzag sequence of images, based on a fixed scale parameter $\alpha$, we can compute the zigzag persistence of the sequence of vector spaces as follows (where $H_p(\mathscr{K})$ denotes $p$-th homology group of $\mathscr{K}$):

$$H_p(\mathscr{K}^{(1)}) \hookrightarrow H_p(\mathscr{K}^{(1)} \cup \mathscr{K}^{(2)}) \hookleftarrow H_p(\mathscr{K}^{(2)}) \hookrightarrow \cdots H_p(\mathscr{K}^{(N-1)} \cup \mathscr{K}^{(N)}) \hookleftarrow H_p(\mathscr{K}^{(N)})$$

The extracted topological information can be summarized in the form of a multiset in $\mathbb{R}^2$, i.e., so-called persistence diagram (ZPD) $Dg_Z = \{p_\rho = (b_\rho, d_\rho) \in \mathbb{R}^2 : d_\rho > b_\rho\} \cup \Delta$ (here $\Delta = \{(\tau, \tau)|\tau \in \mathbb{R}\}$ is the diagonal set containing points counted with infinite multiplicity; including $\Delta$ allows us to compare different ZPDs based on the cost of the optimal matching between their points). For each persistence diagram $Dg_Z$, we compute its zigzag persistence image (ZPI) denoted as $PI_Z$ via the vectorization (denoted as $\text{Vec}(\cdot)$) scheme (Adams et al., 2017; Chen et al., 2021b; 2022; 2023). We provide an exmaple of ZPI generation pipeline in Figure 2, Appendix A. In our experiments, we use both 0- (i.e., connected components) and 1-dimensional (i.e., holes) topological features.

**Theorem 3.2 (Zigzag stability for time–series images)** *Let $\Omega \subset \mathbb{Z}^2$ be finite and $m^{(i)} : \Omega \to \mathbb{R}$ $(i = 1, \ldots, N)$ be grayscale frames. For $\alpha \in \mathbb{R}$ define the lower-star cubical sublevel complexes $\mathscr{K}^{(i)}_\alpha = \{\sigma \subset \Omega \text{ cubical} : \max_{v \in \text{vert}(\sigma)} m^{(i)}(v) \le \alpha\}$ and bridges $\mathscr{K}^{(i,i+1)}_\alpha = \mathscr{K}^{(i)}_\alpha \cup \mathscr{K}^{(i+1)}_\alpha$. Fix $p \ge 0$ and a field $\Bbbk$. For each $\alpha$, let*

$$X_\alpha : \mathscr{K}^{(1)}_\alpha \hookrightarrow \mathscr{K}^{(1,2)}_\alpha \hookleftarrow \mathscr{K}^{(2)}_\alpha \hookrightarrow \cdots \hookleftarrow \mathscr{K}^{(N)}_\alpha, \quad V_\alpha = H_p(X_\alpha; \Bbbk),$$

*and have $\mathbb{V} : \alpha \mapsto V_\alpha$. For another sequence $\tilde{m}^{(i)}$ with $\|m^{(i)} - \tilde{m}^{(i)}\|_\infty \le \varepsilon$ for all $i$, define $\tilde{\mathbb{V}}$ analogously. Let $\text{sh}_\delta$ be the shift $(\text{sh}_\delta \mathbb{V})_\alpha = V_{\alpha+\delta}$. Then there exist natural transformations $\Phi : \mathbb{V} \Rightarrow \text{sh}_\varepsilon \tilde{\mathbb{V}}$ and $\Psi : \tilde{\mathbb{V}} \Rightarrow \text{sh}_\varepsilon \mathbb{V}$ making an $\varepsilon$-interleaving: $(\text{sh}_\varepsilon \Phi) \circ \Psi = \tilde{\eta}$ and $(\text{sh}_\varepsilon \Psi) \circ \Phi = \eta$, where $\eta, \tilde{\eta}$ are the canonical inclusions to the $2\varepsilon$-shift. Hence $d_I(\mathbb{V}, \tilde{\mathbb{V}}) \le \varepsilon$, and (since these modules are pointwise finite-dimensional on a finite grid) their zigzag persistence diagrams satisfy:*

$$d_B\big(\text{ZPD}_p(\mathbb{V}), \text{ZPD}_p(\tilde{\mathbb{V}})\big) \le \varepsilon.$$

The proof of Theorem 3.2 is provided in Appendix B. The theorem suggests that time-zigzag persistence on image sequences is robust to small grayscale fluctuations, so differences observed in the resulting zigzag diagrams reflect genuine structural change rather than noise, which supports reliable comparison across time windows and trustworthy use in downstream analysis.

**Temporal topological learning framework.** To address the challenge of processing multi-scale temporal information, we aggregate multiple time-scale topological features into on one unified representation. More specifically, according to different temporal scales, we use $Q$ different patch lengths $\boldsymbol{P} = \{P_1, P_2, \ldots, P_Q\}$ and obtain the corresponding $Q$ time-series data $\mathbb{M} = \{\boldsymbol{M}_{P_1}, \boldsymbol{M}_{P_2}, \ldots, \boldsymbol{M}_{P_Q}\}$ by using the proposed patching strategy. Given multi-scale time-series image $\mathbb{M}$, we first employ the ZP and vectorization method to generate zigzag persistence images with different scales, and then integrate them into a mixup zigzag persistence image with different coefficients. That is:

$$\boldsymbol{PI}_Z = \sum_{q=1}^{Q} \beta_q PI_Z^{P_q}, \ PI_Z^{P_q} = \text{Vec}(\text{ZP}(\boldsymbol{M}_{P_q})), \tag{3}$$

where $\beta_q$ represent the importance coefficient for $q$-th temporal scale (in this paper, we consider 4 different time scales and hyperparameters $\boldsymbol{\beta} = \{\beta_q\}_{q=1}^{Q}$ are equal to 0.25 (i.e., $\beta_1 = \beta_2 = \beta_3 = \beta_4 = 0.25$)). Next, we apply a two-layer CNN over the mixup zigzag persistence image $\boldsymbol{PI}_Z$, yielding latent dynamic topological representation:

$$\tilde{\boldsymbol{z}} = \text{FC}(\text{LayerNorm}(\text{CNN}(\boldsymbol{PI}_Z))), \tag{4}$$

where LayerNorm denotes the layer normalization to maintain the feature scale, and FC denotes a fully connected layer which flattens convolution results.

### 3.3 MODELING SPATIAL AND TEMPORAL INFORMATION

**Temporal encoding.** Positional encoding is a crucial design in the Transformer architecture for making use of the order of the sequence. To effectively utilize temporal information of STPP, for each event time $t_i$, we map it into the temporal embedding $\boldsymbol{t}_i$ by using a positional encoding (where $D$ denotes the embedding dimension). In summary, we have:

$$[\boldsymbol{t}_i]_j = \begin{cases} \cos\left(\frac{t_i}{10000^{(j-1)/D}}\right), & \text{when } j \text{ is odd,} \\ \sin\left(\frac{t_i}{10000^{(j-1)/D}}\right), & \text{when } j \text{ is even.} \end{cases} \tag{5}$$

For accurate temporal prediction, it is vital to model temporal dependencies, as well as trend shift. Temporal query (TQ) techniques have been successfully applied to learn robust multivariate correlations from multivariate time-series data (Kulkarni et al., 2011; Lin et al., 2025). Inspired by this, in this work, we develop a TQ-aware self-attention module to effectively and adaptively identify temporal patterns inside the sequence of events. Specifically, given the skeleton of the self-attention module, i.e., Self-Attention $= \text{Softmax}(\frac{\boldsymbol{Q}\boldsymbol{K}^\top}{\sqrt{d}})\boldsymbol{V}$ (where $\boldsymbol{Q}$, $\boldsymbol{K}$, and $\boldsymbol{V}$ are queries, keys, and values respectively), we integrate a TQ learnable matrix into the query matrix, and integrate temporal encoding output into both key and value matrices, i.e.,

$$\text{Self-Attention}_{\text{TQ}}(\tilde{\boldsymbol{t}}) = \text{Softmax}\left(\frac{\boldsymbol{Q}_{\text{TQ}}\boldsymbol{K}_{\tilde{\boldsymbol{t}}}^\top}{\sqrt{d_{\tilde{\boldsymbol{t}}}}}\right)\boldsymbol{V}_{\tilde{\boldsymbol{t}}}, \tag{6}$$

where $\boldsymbol{Q}_{\text{TQ}} = \boldsymbol{W}_{\text{TQ}}\boldsymbol{W}^Q$, $\boldsymbol{K}_{\tilde{\boldsymbol{t}}} = \tilde{\boldsymbol{t}}\boldsymbol{W}^K$, $\boldsymbol{V}_{\tilde{\boldsymbol{t}}} = \tilde{\boldsymbol{t}}\boldsymbol{W}^V$, $\boldsymbol{W}_{\text{TQ}}$ is a learnable TQ matrix, $\boldsymbol{W}^Q$, $\boldsymbol{W}^K$, and $\boldsymbol{W}^V$ are projection matrices, and $d_{\tilde{\boldsymbol{t}}}$ is the dimensionality of the queries and the keys. Then the output of the positional encoding, i.e., $\tilde{\boldsymbol{t}} = (\boldsymbol{t}_1, \boldsymbol{t}_2, \ldots, \boldsymbol{t}_K)$ is fed into the self-attention mechanism, and the self-attention output is added back to its input (i.e., $\tilde{\boldsymbol{t}}$) via a residual connection and then normalized with the layer normalization, which stabilizes optimization and gradient flow while preserving the initial embedding as new contextual information is integrated:

$$\tilde{\boldsymbol{o}}^t = \text{LayerNorm}(\tilde{\boldsymbol{t}} + \text{Self-Attention}_{\text{TQ}}(\tilde{\boldsymbol{t}})). \tag{7}$$

**Spatial encoding.** Given the spatial information $g_i$ of event $i$, we apply a lightweight MLP to learn the spatial embedding and we present the output as $\boldsymbol{g}_i$. In our study, the MLP consists of two connected layers with ReLU activation, which are defined as $\boldsymbol{g}_i = \text{Linear}(\text{ReLU}(\text{Linear}(g_i))$.

For spatial encoding, we apply the regular self-attention mechanism over the initial spatial embedding $\tilde{\boldsymbol{g}} = (\boldsymbol{g}_1, \boldsymbol{g}_2, \ldots, \boldsymbol{g}_K)$, yielding the final spatial representation:

$$\tilde{\boldsymbol{o}}^s = \text{LayerNorm}(\tilde{\boldsymbol{g}} + \text{Self-Attention}(\tilde{\boldsymbol{g}})). \tag{8}$$

Following Eqs. 7 and 8, we can obtain latent embeddings in temporal ($\tilde{\boldsymbol{o}}_t$) and spatial ($\tilde{\boldsymbol{o}}_s$) domains separately. However, to jointly learn spatio-temporal representations, the summarization operation is not enough to seamlessly link two domains without an adapter. To integrate spatial and temporal embeddings, next we introduce an unified topology-aware fusion framework.

### 3.4 SPATIO-TEMPORAL FUSION WITH TEMPORAL TOPOLOGY LEARNING FRAMEWORK

The topology-aware fusion framework integrates spatial, temporal, and dynamic topological embeddings, leveraging their complementary strengths to (i) capture spatio-temporal dependencies and (ii) narrow the gap between spatial and temporal domains. The dynamic topology embedding $\tilde{\boldsymbol{z}}$ from TTL encodes spatio-temporal topological patterns and serve as queries in the topology-guided spatio-temporal multi-head attention (TST-MHA) mechanism, while the concatenation of spatial, temporal, and graph embeddings denoted as $\tilde{\boldsymbol{r}} = \oplus(\tilde{\boldsymbol{t}}, \tilde{\boldsymbol{g}}, \boldsymbol{o}_{\mathcal{G}})$ serve as keys and values. The TST-MHA is defined as:

$$\text{TST-MHA}(\boldsymbol{Q}, \boldsymbol{K}, \boldsymbol{V}) = \oplus(\text{head}_1, \ldots, \text{head}_H)\boldsymbol{W}^O, \quad \text{head}_h = \text{Softmax}\left(\frac{\boldsymbol{Q}_h\boldsymbol{K}_h^\top}{\sqrt{d_k}}\right)\boldsymbol{V}_h, \quad (9)$$

where $\boldsymbol{Q}_h = \tilde{\boldsymbol{z}}\boldsymbol{W}^Q$, $\boldsymbol{K}_h = \tilde{\boldsymbol{r}}\boldsymbol{W}^K$, $\boldsymbol{V}_h = \tilde{\boldsymbol{r}}\boldsymbol{W}^V$, and $\boldsymbol{W}_h^Q$, $\boldsymbol{W}_h^K$, $\boldsymbol{W}_h^V$, and $\boldsymbol{W}^O$ are learnable projection matrices. $d_k$ denotes the head dimension, and $H$ is the number of attention heads. The proposed fusion framework integrates spatial, temporal, and topology-aware dynamic information,

capturing both local and global dependencies. We derive the Lipschitz bound for TST-MHA, see Theorem B.1 in Appendix B.

Finally, the combined embedding (i.e., from spatial domain, temporal domain, and TST-MHA) will be transformed by the feedforward layer, which is formally computed as:

$$\tilde{\boldsymbol{o}}^{TST} = \text{Feed-Forward}(\text{LayerNorm}(\tilde{\boldsymbol{r}} + \text{TST-MHA}(\boldsymbol{Q}, \boldsymbol{K}, \boldsymbol{V}))). \tag{10}$$

## 3.5 SPATIO-TEMPORAL FORWARD AND REVERSE DIFFUSION

For each event $\boldsymbol{x}_i = (\tau_i, g_i)$ in the sequence (where $\tau_i$ is the time interval since the last event), we perform the forward diffusion process as a Markov process over the spatial and temporal dimensions as $(\boldsymbol{x}_i^0, \boldsymbol{x}_i^1, ..., \boldsymbol{x}_i^K)$, where $K$ is the number of diffusion steps. That is, we add small amount of Gaussian noise step by step to the space and time values until they are close to pure Gaussian noises. The forward process of our diffusion model on spatial and temporal dimensions can be written as:

$$q_{st}(\boldsymbol{x}_i^k|\boldsymbol{x}_i^{k-1}) = (q(\tau_i^k|\tau_i^{k-1}), q(g_i^k|g_i^{k-1})), \tag{11}$$

where the recursive formula is $q(\boldsymbol{x}^k|\boldsymbol{x}^{k-1}) = \mathcal{N}(\boldsymbol{x}^k; \sqrt{1-\beta_k}\boldsymbol{x}^k, \beta_k\boldsymbol{I})$, $\mathcal{N}(\cdot, \cdot)$ denotes the Gaussian distribution used to generate the noise, $\boldsymbol{I}$ is the identity matrix, and $\overline{\alpha}_k = \prod_{i=1}^{k}(1-\beta_i)$ (where $\beta_{1:K} \in (0, 1)$). The purpose of the reverse process is enable our diffusion model to learn the denoising ability of noisy spatial and temporal information. Specifically, we aim to reconstruct the point $\boldsymbol{x}_i = (\tau_i, g_i)$ with the learned model over $K$ steps $\boldsymbol{x}_i^K \to \boldsymbol{x}_i^{K-1} \to \cdots \to \boldsymbol{x}_i^0$. Further, we also incorporate latent spatio-temporal embedding denoted as $\tilde{\boldsymbol{o}}_{i-1}$ (where $\tilde{\boldsymbol{o}}_{i-1} = \oplus(\tilde{\boldsymbol{o}}_{i-1}^s, \tilde{\boldsymbol{o}}_{i-1}^g, \tilde{\boldsymbol{o}}_{i-1}^{TST})$) into the backward diffusion process which helps to guide the denoising process towards the clean sample. The denoising transition step is outlined as follows:

$$p_\theta(\boldsymbol{x}_i^{k-1}|\boldsymbol{x}_i^k, \tilde{\boldsymbol{o}}_{i-1}) = p_\theta(\tau_i^{k-1}|\tau_i^k, g_i^k, \tilde{\boldsymbol{o}}_{i-1})p_\theta(g_i^{k-1}|\tau_i^k, g_i^k, \tilde{\boldsymbol{o}}_{i-1}). \tag{12}$$

In our experiments, we employ the cross-attentive conditional denoising decoder (Wang et al., 2024) which incorporates predicted values $\tau_i^{k+1}$, $g_i^{k+1}$ in temporal and spatial dimensions respectively, graph learning output $\boldsymbol{o}_{\mathcal{G}}$ (see Eq. 2), and denoising step $k$ with positional encoding and leverages the latent spatio-temporal embedding $\tilde{\boldsymbol{o}}_{i-1}$ for the guidance in the conditional denoising process. To predict future event, we utilize the inference framework proposed by Yuan et al. (2023a).

# 4 EXPERIMENTS

## 4.1 EXPERIMENT SETTINGS

**Datasets.** In our experiments, we use 5 real-world datasets, i.e., **JPN Earthquake:** Earthquake with a magnitude of at least 2.5 in Japan from 1990 to 2020; **COVID-19:** COVID-19 dataset is collected from publicly released COVID19 cases in New Jersey state from March 2020 to July 2020; **US Earthquake:** The US earthquake dataset contains earthquake occurrences from December 2023 to January 2024 in US; **Theft:** The theft data is collected by the Philadelphia police department from January 2025 to April 2025 in Philadelphia; **311 Service:** Similar to theft data, we collect Philadelphia 311 service dataset from OpenDataPhilly from January 2025 to June 2025. More details of data resources and train/validation/test split ratio are in Appendix A.

**Baselines and Evaluation Protocol.** We compare TEN-DM with 17 baselines, including 3 SPP baselines, 10 TPP baselines, and 4 STPP baselines. **SPP baselines:** (i) Conditional Kernel Density Estimator (KDE) (Rosenblatt, 1969; Zhao & Tabak, 2025); (ii) Continuous Normalizing Flow (CNF) (Chen et al., 2018); and (iii) Time-Varing Continuous Normalizing Flow (TVCNF) (Chen et al., 2021a). **TPP Baselines:** (i) homogeneous Poisson process (Kingman, 1992); (ii) Hawkes Process (Hawkes, 1971); (iii) Self-correcting process (Isham & Westcott, 1979); (iv) Recurrent Marked Temporal Point Process (RMTPP) (Du et al., 2016); (v) Neural Hawkes Process (NHP) (Mei & Eisner, 2017); (vi) Transformer Hawkes Process (THP) (Zuo et al., 2020); (vii) Self-Attentive Hawkes Process (SAHP) (Zhang et al., 2020); (viii) Log Normal Mixture model (LogNormMix) (Shchur et al., 2020); (ix) Wasserstein GAN (WGAN) (Xiao et al., 2017); and (x) Neural Jump-Diffusion Temporal Point Process (NJDTPP) (Zhang et al., 2024). **STPP Baselines:** (i) Neural Jump Stochastic Differential Equations (NJSDE) (Jia & Benson, 2019); (ii) Neural Spatio-temporal Point Process

(NSTPP) (Chen et al., 2021a); (iii) Deep Spatio-temporal Point Process (DeepSTPP) (Zhou et al., 2022); and (iv) Spatio-temporal Diffusion Point Processes (DSTPP) (Yuan et al., 2023b). We have included a detailed introduction about baselines in Appendix A. We evaluate prediction on the next event in both space and time. The spatial error is measured by the Euclidean distance between the predicted and ground truth location, and the temporal error by root-mean-square error (RMSE) between the predicted and ground truth time interval.

**Implementation Details.** We run our experiments on 4 NVIDIA RTX A5000 GPU cards with 24GB memory. Optimization uses AdamW($\beta_1 = 0.9, \beta_2 = 0.99$) and a learning rate warm-up from 0 to a peak selected from $\{1e^{-3}, 3e^{-4}\}$ followed by linear decay to $5e^{-5}$ for 1000 epochs. More details can be found in the Appendix A. Evaluation metrics report spatial Euclidean distance and temporal RMSE on 3 runs with different random seeds. Code is available at https://github.com/yl3564/TEN-DM. All datasets used are publicly available; we include raw data resources, and the train/val/test splits used in our experiments. Hyperparameter settings are reported in "Implementation Details".

## 4.2 RESULTS

Table 1 compares the forecasting errors of TEN-DM with 17 baseline models across 5 real-world datasets in spatial and temporal domains. Lower Euclidean distance and RMSE indicate higher forecasting accuracy. As shown in Table 1, TEN-DM significantly outperforms all baselines across all 5 datasets in both Euclidean distance and RMSE except for JPN earthquake data in temporal dimension. Notably, on COVID-19, US earthquake, and 311 service, our TEN-DM is statistically significantly (with $p$-value $< 0.05$) better than runner-ups in both spatial and temporal domains. **Superior performance against SPP baselines.** TEN-DM outperforms all 3 baselines; compared with SPP runner-ups, TEN-DM achieves 21.97%, 42.97%, 9.37%, 0.29%, and 0.55% relative improvement on JPN earthquake, COVID-19, US earthquake, theft, and 311 service datasets respectively. **Superior performance against TPP baselines.** TEN-DM statistically significantly outperforms all TPP baselines with 6.74%, 42.53%, 57.14%, 38.02% and 13.89% relative improvements on JPN earthquake, COVID-19, US earthquake, theft, and 311 service datasets respectively (compared with runner-ups). **Better performance against STPP baselines.** Compared with the runner-up (i.e., DSTPP), TEN-DM achieves 7.16% and 6.90% relative improvement on COVID-19 dataset in spatial and temporal dimensions respectively; a 17.08% and 3.66% relative improvement on theft and 311 service datasets respectively in temporal dimension. Computational complexity and running time are provided in Appendix A.

Table 1: Performance evaluation for predicting both time and space of the next event. We use Euclidean distance and RMSE to predict errors of the spatial domain and temporal domain respectively. Here * denotes $p$-value $< 0.05$ (i.e., statistically significant results).

| Model | JPN Earthquake | | COVID-19 | | US Earthquake | | Theft | | 311 Service | |
|---|---|---|---|---|---|---|---|---|---|---|
| | Spatial ↓ | Temporal ↓ | Spatial ↓ | Temporal ↓ | Spatial ↓ | Temporal ↓ | Spatial ↓ | Temporal ↓ | Spatial ↓ | Temporal ↓ |
| Conditional KDE | 11.300±0.658 | - | 0.688±0.047 | - | 41.999±0.036 | - | 0.073±0.000 | - | 0.057±0.001 | - |
| CNF | 8.480±0.054 | - | 0.559±0.000 | - | 42.634±0.036 | - | 0.072±0.000 | - | 0.056±0.000 | - |
| TVCNF | 8.110±0.001 | - | 0.560±0.000 | - | 42.155±2.122 | - | 0.072±0.000 | - | 0.056±0.000 | - |
| Poisson | - | 0.631±0.017 | - | 0.463±0.021 | - | 0.431±0.035 | - | 0.626±0.016 | - | 1.259±0.032 |
| Hawkes | - | 0.544±0.010 | - | 0.672±0.088 | - | 0.121±0.002 | - | 0.629±0.027 | - | 1.486±0.024 |
| Self-correcting | - | 11.200±0.486 | - | 2.830±0.141 | - | 3.130±0.346 | - | 0.659±0.024 | - | 2.526±0.122 |
| RMTPP | - | 0.424±0.009 | - | 1.320±0.024 | - | 1.626±0.030 | - | 0.583±0.028 | - | 1.742±0.009 |
| NHP | - | 1.860±0.023 | - | 2.130±0.100 | - | 3.749±0.153 | - | 0.612±0.021 | - | 2.314±0.065 |
| THP | - | 2.440±0.021 | - | 0.611±0.008 | - | 1.242±0.009 | - | 0.527±0.017 | - | 0.976±0.051 |
| SAHP | - | 0.409±0.002 | - | 0.184±0.024 | - | 0.457±0.008 | - | 0.694±0.039 | - | 1.128±0.087 |
| LogNormMix | - | 0.593±0.005 | - | 0.168±0.011 | - | 0.474±0.062 | - | 0.501±0.008 | - | 2.675±0.009 |
| WGAN | - | 0.481±0.007 | - | 0.124±0.002 | - | 0.766±0.001 | - | 0.699±0.019 | - | 2.083±0.093 |
| NJDTPP | - | 0.396±0.003 | - | 0.790±0.098 | - | 0.535±0.110 | - | 0.508±0.046 | - | 0.902±0.012 |
| NJSDE | 9.980±0.024 | 0.465±0.009 | 0.641±0.009 | 0.137±0.001 | 51.784±0.013 | 0.081±0.000 | 0.099±0.002 | 0.465±0.006 | 0.067±0.001 | 0.865±0.033 |
| NSTPP | 8.110±0.000 | 0.547±0.010 | 0.560±0.000 | 0.145±0.000 | 59.833±0.006 | 0.102±0.000 | 0.097±0.000 | 0.534±0.046 | 0.072±0.000 | 0.870±0.027 |
| DeepSTPP | 9.200±0.000 | *0.341±0.000 | 0.687±0.000 | 0.197±0.000 | 56.322±0.178 | 0.093±0.000 | 0.089±0.001 | 0.420±0.007 | 0.059±0.000 | 0.830±0.035 |
| DSTPP | 6.770±0.000 | 0.375±0.000 | 0.419±0.000 | 0.093±0.000 | 38.892±0.104 | 0.078±0.000 | 0.0701±0.0001 | 0.425±0.002 | 0.0551±0.0001 | 0.821±0.001 |
| **TEN-DM (ours)** | *6.649±0.041 | 0.371±0.003 | *0.391±0.001 | *0.087±0.001 | *38.543±0.200 | *0.077±0.000 | 0.0700±0.0001 | *0.363±0.017 | *0.0547±0.0002 | *0.792±0.026 |

## 4.3 ABLATION STUDIES

To examine the effectiveness of the proposed components, we conduct experiments without graph learning (i.e., W/o Graph), TQ-SA (i.e., W/o TQ-SA), and TTL (i.e., W/o TTL) on JPN earthquake, COVID-19, 311 service datasets. From Table 2 in Appendix A, the results indicate that employing graph learning, TQ-SA, and TTL significantly improves model performance which demonstrate the

effectiveness of the TEN-DM model architecture. For instance, (i) removing graph learning leads to Euclidean distance 0.24% and 0.26% increases on JPN earthquake and COVID-19 datasets respectively; (ii) removing TQ-SA leads to Euclidean distance 1.28% increases on COVID-19; (iii) removing TTL severely limits TEN-DM's ability to capture spatio-temporal topological information, i.e., leading to RMSE 4.60% and 3.03% increases on COVID-19 and 311 service datasets respectively.

## 5   CONCLUSION

We present TEN-DM, a novel diffusion model that leverages geometric and topological learning frameworks to capture dynamic local and global spatio-temporal dependencies for STPP forecasting. Integration of the GCL, TQ-SA, and TTL modules enhances the diffusion model's ability to learn complex spatio-temporal interactions, periodic patterns, and local and global dynamic topological information. Extensive experiments on 5 real-world datasets demonstrated the effectiveness of TEN-DM, achieving state-of-the-art performance across diverse datasets and dimensions. Our work establishes a new direction in STPP forecasting by highlighting the potential of geometric and topological DL in capturing intricate temporal and spatial dependencies.

## ACKNOWLEDGMENTS

The authors would like to express their gratitude to the AC and Reviewers for providing constructive and thought provoking feedback. This work was supported in part by NSF Awards TIP/ITE-2333703, OAC-2530471/2530469 and DMS 2533984/2533985, as well as by the U.S. Department of Energy (DOE), Office of Science, Advanced Scientific Computing Research (ASCR) program under the Scientific Discovery through Advanced Computing (SciDAC) Institute "LEADS: LEarning-Accelerated Domain Science".

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

## THE USE OF LARGE LANGUAGE MODELS (LLMs)

We only use large language models (LLMs) for rephrasing sentences and correcting grammar. It is important to note that the LLMs are not involved in the data analysis, research methodology, or experimental design.

## A  MORE DETAILS

**Cubical complex.** An elementary interval is a closed subset of $\mathbb{R}$ of the form $[z, z+1]$ with $z \in \mathbb{Z}$, together with the degenerate interval $[z, z]$ (i.e., a point). An elementary cube is any Cartesian product

$$Q = I_1 \times \cdots \times I_d,$$

where $d$ is the dimension of the space; for each $j$ we have $I_j = [z_j, z_j + 1]$ with $z_j \in \mathbb{Z}$. The dimension of $Q$ is:

$$\dim Q = \#\{ j \in \{1, \ldots, d\} : I_j \text{ is nondegenerate } ([z_j, z_j + 1]) \}.$$

A cubical complex $\mathscr{K}$ is a finite collection of such axis-aligned elementary cubes in some $\mathbb{R}^d$ that satisfies: if $Q \in \mathscr{K}$ and $F$ is obtained from $Q$ by replacing one or more nondegenerate factors $[z_j, z_j + 1]$ with an endpoint $\{z_j\}$ or $\{z_j + 1\}$, then $F \in \mathscr{K}$; for any $Q, Q' \in \mathscr{K}$, the intersection $Q \cap Q'$ is either empty or a (possibly degenerate) common face of both.

**Persistent homology.** By utilizing a multi-scale approach to shape description, PH addresses the intrinsic limitations of classical homology and allows for the retrieval of shape patterns that tend to persist over multiple scales and, hence, are likelier to play an important role for a given downstream task. The main idea is to select some suitable scale parameters $\alpha$ and then to assess changes in shape (or more formally homology) that occur to an image $m$, which evolves with respect to $\alpha$. Specifically, let $f$ be a filtration function that maps every simplex to the maximum function value of its vertices (in our case the grayscale value) and let $m_\alpha = f^{-1}((-\infty, \alpha]), \alpha \in \mathbb{R}$. Setting an increasing sequence of (dis)similarity thresholds $\alpha$, i.e., $\alpha_0 < \alpha_1 < \cdots < \alpha_h$, sub-images are generated in a nested sequence of cubical complexes (and their connected components, loops, and voids are recorded), i.e., $m_{\alpha_0} \subset m_{\alpha_1} \subset \cdots \subset m_{\alpha_h}$ and we can construct the corresponding sequence of complexes, i.e., $\mathscr{K}_{\alpha_0} \subset \mathscr{K}_{\alpha_1} \subset \cdots \subset \mathscr{K}_{\alpha_h}$ which are referred to as the lower-star filtered cubical complexes. Based on the evolution of these simplicial complexes through a sequence of thresholds, the homology groups are induced as $\{H_p(\mathscr{K}_{\alpha_0}), H_p(\mathscr{K}_{\alpha_1}), \ldots, H_p(\mathscr{K}_{\alpha_h})\}$, where $H_p(\mathscr{K}_{\alpha_j})$ represents the $p$-th homology group of $\mathscr{K}_{\alpha_j}$.

**Zigzag persistence image generation pipeline.** Figure 2 illustrate zigzag persistence image generation pipeline.

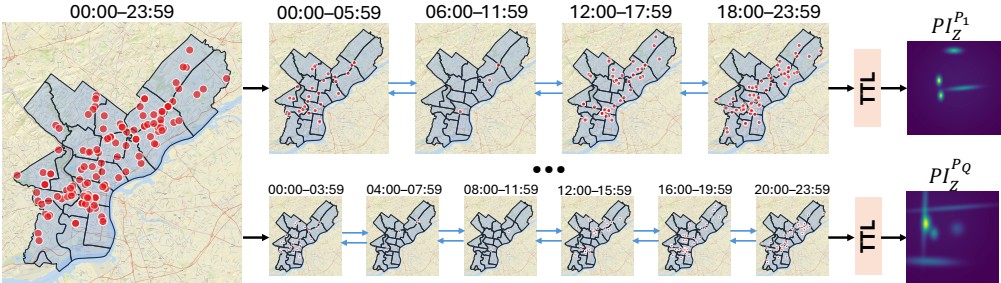

Figure 2: Pipeline for generating ZPI with different time scales.

**Datasets.** In our experiments, we use five real-world STPP datasets, i.e., **Japan Earthquake:** Earthquake with a magnitude of at least 2.5 in Japan from 1990 to 2020 were sourced from the U.S. Geological Survey[1]. It contains 91,897 events in total. Sequences are formed using sliding windows of 30 days. The dataset is partitioned into nonoverlapping splits containing 950 training sequences, 50 validation sequences, and 50 test sequences. Sequence lengths range from 22 to 554. **COVID-19:**

---

[1]https://earthquake.usgs.gov/earthquakes/search/

COVID-19 dataset is collected from publicly released COVID19 cases in New Jersey state from March 2020 to July 2020 by The New York Times[2]. This dataset includes 161,307 recorded cases, aggregated at the county level with a 7-day window size. It is split without overlap into 1450 training sequences, 100 validation sequences, and 100 test sequences. Sequence lengths range from 5 to 287. **US Earthquake:** The US earthquake dataset[3] contains 9,451 earthquake occurrences from 12/27/2023 to 01/26/2024 in US, and is divided into 21 training sequences, 5 validation sequences, and 5 testing sequences. Sequences lengths range from 5 to 512. **Theft:** The theft data is collected by the Philadelphia police department (PPD)[4] and we select 11,405 cases from the most recent theft incidents from 01/01/2025 - 04/30/2025 in Philadelphia. We use 24-hour sliding windows to form the dataset that contains 100 training sequences, 9 validation sequences, and 9 testing sequences. Sequence lengths range from 6 to 138. **311 Service:** Similar to theft data, we collect Philadelphia 311 service dataset from OpenDataPhilly. In this work, we focus on 9,597 illegal dumping reports from 01/01/2025 - 06/30/2025, and split the dataset into disjoint training, validation, and test sets with 151, 15, and 15 sequences respectively. Sequence lengths range from 9 to 111.

**Baselines:**

- **SPP Baselines:** We evaluate methods that model continuous spatial density function. A learned parameterized conditional kernel density estimator (KDE) (Rosenblatt, 1969) that models $p(x|t)$ as a Gaussion mixture model conditioned on historical events. Continuous normalizing flow (CNF) (Chen et al., 2018) defines the invertible flow as a neural ODE and learns a continuous probability density over space. Time-varing Continuous normalizing flow (TVCNF) (Chen et al., 2021a) extends CNF by making the flow dynamics dependent on timestamps.

- **TPP Baselines:** Homogeneous Poisson process (Kingman, 1992) models the probability of an event as proportional to the time interval length. Hawkes Process (Hawkes, 1971) is a self-exciting process where the historical occurrence of events can positively influences the probability of future event occurrence. Self-correcting process (Isham & Westcott, 1979) is opposite to self-exciting processes, which means historical occurrence of events can decrease the probability of future event occurrence and the intensity is negatively influenced upon a new event happened. The Recurrent Marked Temporal Point Process (RMTPP) (Du et al., 2016) simultaneously model the event timings and the markers by viewing the intensity function as a nonlinear one and apply a recurrent neural network to embed the event history. Neural Hawkes Process (NHP) (Mei & Eisner, 2017) uses neurally self-modulating multivariate point process where the intensities of each event type change by a novel LSTM. Transformer Hawkes Process (THP) (Zuo et al., 2020) replaces RNNs with self-attention mechanism to better capture long-term dependencies and keeps computational efficiency. Self-Attentive Hawkes Process (SAHP) (Zhang et al., 2020) also uses self-attention mechanism and modify positional encoding so that time intervals become phase shifts in sinusoidal encoding, which improves the usual "order-only" encoding. Log Normal Mixture model (LogNormMix) (Shchur et al., 2020) learns probability density by a log-normal mixture model. Wasserstein GAN (WGAN) (Xiao et al., 2017) transforms nuisance processes to a target one, providing an intensity-free approach for point processes modeling. Neural Jump-Diffusion Temporal Point Process (NJDTPP) (Zhang et al., 2024) which formulates a neural jump–diffusion SDE with neural parameterizations of the drift, diffusion, and jump coefficient functions.

- **STPP Baselines:** Neural Jump Stochastic Differential Equations (NJSDE) (Jia & Benson, 2019) learns hybrid continuous–discrete dynamics and generates a piecewise-continuous latent trajectory to model the temporal point processes. The spatial distribution is modeled with a Gaussian mixture model. Neural Spatio-temporal Point Process (NSTPP) (Chen et al., 2021a) proposed two novel neural architectures which adds event-time updates and attention for long histories non-trivially. Deep Spatio-temporal Point Process (DeepSTPP) (Zhou et al., 2022) proposes a nonparametric space–time intensity governed by a neural latent process. Spatio-temporal Diffusion Point Processes (DSTPP) (Yuan et al., 2023b) uses conditional diffusion that learns the joint distribution of next event's time and location.

---

[2]https://github.com/nytimes/covid-19-data

[3]https://earthquake.usgs.gov/earthquakes/feed/v1.0/csv.php

[4]http://opendataphilly.org/

**Implementation Details.** We run our experiments on 4 NVIDIA RTX A5000 GPU cards with 24GB memory. Optimization uses AdamW($\beta_1 = 0.9, \beta_2 = 0.99$) and a learning rate warm-up from 0 to a peak selected from $\{1e^{-3}, 3e^{-4}\}$ followed by linear decay to $5e^{-5}$ for 1000 epochs. Training and sampling timesteps are selected from $\{200, 500\}$. We tune the batch size over $\{32, 64\}$. We consider three losses applied to the diffusion objective: $\ell_1$ loss, $\ell_2$ loss, and Euclidean loss. The length of the learnable TQ vectors is selected from $\{7, 24, 30\}$. The number of attention heads is selected from $\{2, 3, 4\}$. For ZPI, we set the grid size to be $50 \times 50$. For graph pretraining, we learn per-sequence graph embeddings with a graph auto-encoder (GAE) (Kipf & Welling, 2016) with graph attention network (GAT) (Veličković et al., 2018) encoder trained on each graph independently. Optimization uses Adam with a learning rate of 0.01 over 400 epochs. Evaluation metrics report spatial Euclidean distance and temporal RMSE on 3 runs with different random seeds.

**Ablation studies.**

Table 2: Ablation studies.

| Model | JPN Earthquake | | COVID-19 | | 311 Service | | Theft | |
|---|---|---|---|---|---|---|---|---|
| | Spatial ↓ | Temporal ↓ | Spatial ↓ | Temporal ↓ | Spatial ↓ | Temporal ↓ | Spatial ↓ | Temporal ↓ |
| **TEN-DM** | **6.649±0.041** | **0.371±0.003** | **0.391±0.001** | **0.087±0.001** | **0.0547±0.0002** | **0.792±0.026** | **0.0700±0.0001** | **0.363±0.017** |
| W/o Graph | 6.665±0.054 | 0.372±0.000 | 0.392±0.001 | 0.088±0.001 | 0.0547±0.0003 | 0.798±0.004 | 0.0701±0.0001 | 0.391±0.015 |
| W/o TQ-SA | 6.663±0.031 | 0.373±0.001 | 0.396±0.004 | 0.088±0.000 | 0.0548±0.0001 | 0.807±0.009 | 0.0702±0.0001 | 0.374±0.020 |
| W/o TTL | 6.663±0.027 | 0.372±0.000 | 0.405±0.003 | 0.091±0.001 | 0.0549±0.0002 | 0.816±0.004 | 0.0701±0.0001 | 0.425±0.001 |

**Computational complexity.** The computational complexity of GNN with $l$ layers is $\mathcal{O}(\xi|\mathcal{E}|\sum_{i=1}^{l} d_i)$ where $\xi$ denotes the total number of gradient descent. The computational complexity of cubical zigzag persistence for connected components and is $\mathcal{O}(U \log^2 n + U \log U)$ and $\mathcal{O}(U \log^2 n + U \log U + n \log n)$ where $U = \sum_{t=1}^{T} \Delta_t$, $n$ denotes pixels per frame, and $T$ is the number of frames. As shown in Table 3, we also report the running time (training time per epoch) of our TEN-DM model on all 5 datasets.

Table 3: Running time (in seconds (s)) per epoch.

| Model | JPN Earthquake | COVID-19 | US Earthquake | Theft | 311 Service |
|---|---|---|---|---|---|
| TEN-DM | 14.096 | 22.873 | 1.846 | 5.982 | 4.570 |

# B PROOFS

**Proof of Theorem 3.2.**

*Proof:* The bound $\|m^{(i)} - \tilde{m}^{(i)}\|_\infty \le \varepsilon$ implies, for every $i$ and $\alpha$,

$$\mathcal{K}_\alpha^{(i)} \hookrightarrow \tilde{\mathcal{K}}_{\alpha+\varepsilon}^{(i)}, \quad \tilde{\mathcal{K}}_\alpha^{(i)} \hookrightarrow \mathcal{K}_{\alpha+\varepsilon}^{(i)},$$

and likewise for bridge nodes: $\mathcal{K}_\alpha^{(i,i+1)} \hookrightarrow \tilde{\mathcal{K}}_{\alpha+\varepsilon}^{(i,i+1)}$ and the reverse inclusion with tildes and non-tildes swapped. Objectwise, these inclusions assemble into morphisms of zigzags

$$I_\alpha : X_\alpha \to \tilde{X}_{\alpha+\varepsilon}, \quad J_\alpha : \tilde{X}_\alpha \to X_{\alpha+\varepsilon},$$

which commute with all internal arrows. Moreover, on each node,

$$\tilde{\mathcal{K}}_\alpha \hookrightarrow \mathcal{K}_{\alpha+\varepsilon} \hookrightarrow \tilde{\mathcal{K}}_{\alpha+2\varepsilon}, \quad \mathcal{K}_\alpha \hookrightarrow \tilde{\mathcal{K}}_{\alpha+\varepsilon} \hookrightarrow \mathcal{K}_{\alpha+2\varepsilon},$$

so $(\mathrm{sh}_\varepsilon I)_\alpha \circ J_\alpha = \tilde{\iota}_\alpha$ and $(\mathrm{sh}_\varepsilon J)_\alpha \circ I_\alpha = \iota_\alpha$, the canonical $2\varepsilon$-shift inclusions of zigzags. Applying $H_p(-; \Bbbk)$ yields natural transformations

$$\Phi_\alpha = H_p(I_\alpha) : V_\alpha \to \tilde{V}_{\alpha+\varepsilon}, \qquad \Psi_\alpha = H_p(J_\alpha) : \tilde{V}_\alpha \to V_{\alpha+\varepsilon},$$

satisfying $(\mathrm{sh}_\varepsilon \Phi)_\alpha \circ \Psi_\alpha = \tilde{\eta}_\alpha$ and $(\mathrm{sh}_\varepsilon \Psi)_\alpha \circ \Phi_\alpha = \eta_\alpha$. Hence $V$ and $\tilde{V}$ are $\varepsilon$-interleaved, so $d_I(V, \tilde{V}) \le \varepsilon$. Since $\Omega$ is finite, the modules are pointwise finite-dimensional and constructible; algebraic stability for zigzag persistence then gives $d_B(\mathrm{ZPD}_p(V), \mathrm{ZPD}_p(\tilde{V})) \le d_I(V, \tilde{V}) \le \varepsilon$. ∎

**Theorem B.1 (Lipschitz bound for topology-guided multi-head attention)** *Let $\tilde{z}$ be topology embeddings and $\tilde{r}$ be concatenated spatial, temporal, and graph embeddings. Consider a single multi-head attention block with $h$ heads,*

$$\text{TST-MHA}(\tilde{z}, \tilde{r}) = \oplus_h \left( \text{Softmax}\left( \frac{\boldsymbol{Q}_h(\tilde{z})\,\boldsymbol{K}_h(\tilde{r})^\top}{\sqrt{d_k}} \right) \boldsymbol{V}_h(\tilde{r}) \right) \boldsymbol{W}^O,$$

*where $\boldsymbol{Q}_h, \boldsymbol{K}_h, \boldsymbol{V}_h, \boldsymbol{W}^O$ are linear maps with operator norms bounded by $M_Q, M_K, M_V, M_O$, and logits are uniformly bounded by $B$ (e.g., via clipping or LayerNorm). Then the map $(\tilde{z}, \tilde{r}) \mapsto \text{TST-MHA}(\tilde{z}, \tilde{r})$ is jointly Lipschitz:*

$$\left\| \text{TST-MHA}(\tilde{z}, \tilde{r}) - \text{TST-MHA}(\tilde{z}', \tilde{r}') \right\|_F \leq \mathscr{L}\left( \|\tilde{z} - \tilde{z}'\|_F + \|\tilde{r} - \tilde{r}'\|_F \right),$$

*with $\mathscr{L} = c(h, B)\, M_O\,(M_Q + M_K + M_V)$ for an explicit $c(h, B)$ depending only on the number of heads and the softmax Lipschitz constant on a $B$–bounded domain.*

*Proof:* For head $h \in \{1, \dots, H\}$ define

$$\boldsymbol{L}_h(\tilde{z}, \tilde{r}) = \frac{\boldsymbol{Q}_h(\tilde{z})\,\boldsymbol{K}_h(\tilde{r})^\top}{\sqrt{d_k}}, \quad \boldsymbol{S}_h(\tilde{z}, \tilde{r}) = \text{Softmax}\big(\boldsymbol{L}_h(\tilde{z}, \tilde{r})\big), \quad \boldsymbol{P}_h(\tilde{z}, \tilde{r}) = \boldsymbol{S}_h(\tilde{z}, \tilde{r})\,\boldsymbol{V}_h(\tilde{r}).$$

Then $\text{TST-MHA}(\tilde{z}, \tilde{r}) = \big( \oplus_{h=1}^H \boldsymbol{P}_h(\tilde{z}, \tilde{r})\big)W^O$. Let $\boldsymbol{S}_h' = \boldsymbol{S}_h(\tilde{z}', \tilde{r}')$, $\boldsymbol{P}_h' = \boldsymbol{P}_h(\tilde{z}', \tilde{r}')$. By addition/subtraction,

$$\boldsymbol{P}_h - \boldsymbol{P}_h' = (\boldsymbol{S}_h - \boldsymbol{S}_h')\,\boldsymbol{V}_h(r') + \boldsymbol{S}_h\big(\boldsymbol{V}_h(r) - \boldsymbol{V}_h(r')\big).$$

Assume the logits are uniformly bounded, i.e., $\|\boldsymbol{L}_h(\tilde{z}, \tilde{r})\|_\infty, \|\boldsymbol{L}_h(\tilde{z}', \tilde{r}')\|_\infty \leq B$ (e.g., via Layer-Norm/clipping). The row-wise softmax is Lipschitz on this $B$–bounded set: for some $\mathscr{L}_{\text{soft}}(B) > 0$,

$$\left\| \boldsymbol{S}_h - \boldsymbol{S}_h' \right\|_F \leq \mathscr{L}_{\text{soft}}(B) \left\| \boldsymbol{L}_h(\tilde{z}, \tilde{r}) - \boldsymbol{L}_h(\tilde{z}', \tilde{r}') \right\|_F.$$

Moreover each $\boldsymbol{S}_h$ is row–stochastic with entries controlled by $B$, hence $\|\boldsymbol{S}_h\|_2 \leq C_S(B)$ for some $C_S(B)$.

With operator-norm bounds $\|\boldsymbol{Q}_h\|_{\text{op}} \leq M_Q$, $\|\boldsymbol{K}_h\|_{\text{op}} \leq M_K$, $\|\boldsymbol{V}_h\|_{\text{op}} \leq M_V$ and submultiplicativity,

$$\left\| \boldsymbol{L}_h(\tilde{z}, \tilde{r}) - \boldsymbol{L}_h(\tilde{z}', \tilde{r}') \right\|_F \leq \frac{M_Q M_K}{\sqrt{d_k}} \left( \|\tilde{z} - \tilde{z}'\|_F + \|\tilde{r} - \tilde{r}'\|_F \right), \quad \left\| \boldsymbol{V}_h(\tilde{r}) - \boldsymbol{V}_h(\tilde{r}') \right\|_F \leq M_V \|\tilde{r} - \tilde{r}'\|_F.$$

Also $\|\boldsymbol{V}_h(\tilde{r}')\|_2 \leq c(B)\, M_V$ for a harmless constant $c(B)$ (absorbing input–norm control due to normalization).

Applying these bounds to $\boldsymbol{P}_h - \boldsymbol{P}_h' = (\boldsymbol{S}_h - \boldsymbol{S}_h')\,\boldsymbol{V}_h(\tilde{r}') + \boldsymbol{S}_h\big(\boldsymbol{V}_h(\tilde{r}) - \boldsymbol{V}_h(\tilde{r}')\big)$,

$$\left\| \boldsymbol{P}_h - \boldsymbol{P}_h' \right\|_F \leq \left\| \boldsymbol{S}_h - \boldsymbol{S}_h' \right\|_F \left\| \boldsymbol{V}_h(\tilde{r}') \right\|_2 + \left\| \boldsymbol{S}_h \right\|_2 \left\| \boldsymbol{V}_h(\tilde{r}) - \boldsymbol{V}_h(\tilde{r}') \right\|_F$$

$$\leq c(B)\, M_V\, L_{\text{soft}}(B) \frac{M_Q M_K}{\sqrt{d_k}} \left( \|\tilde{z} - \tilde{z}'\|_F + \|\tilde{r} - \tilde{r}'\|_F \right) + C_S(B)\, M_V\, \|\tilde{r} - \tilde{r}'\|_F.$$

Hence, for a constant $c_1(B)$ depending only on $B$,

$$\left\| \boldsymbol{P}_h - \boldsymbol{P}_h' \right\|_F \leq c_1(B) \left( M_Q M_K M_V + M_V \right) \left( \|\tilde{z} - \tilde{z}'\|_F + \|\tilde{r} - \tilde{r}'\|_F \right).$$

Concatenation across heads gives

$$\left\| \oplus_{h=1}^H \big( \boldsymbol{P}_h - \boldsymbol{P}_h' \big) \right\|_F \leq \sqrt{H} \max_h \left\| \boldsymbol{P}_h - \boldsymbol{P}_h' \right\|_F \leq \sqrt{H}\, c_1(B) \left( M_Q M_K M_V + M_V \right) \left( \|\tilde{z} - \tilde{z}'\|_F + \|\tilde{r} - \tilde{r}'\|_F \right).$$

Finally, multiplying by $\boldsymbol{W}^O$ with $\|\boldsymbol{W}^O\|_{\text{op}} \leq M_O$,

$$\left\| \text{TST-MHA}(\tilde{z}, \tilde{r}) - \text{TST-MHA}(\tilde{z}', \tilde{r}') \right\|_F \leq M_O\, \sqrt{H}\, c_1(B) \left( M_Q M_K M_V + M_V \right) \left( \|\tilde{z} - \tilde{z}'\|_F + \|\tilde{r} - \tilde{r}'\|_F \right).$$

Using AM–GM to bound $M_Q M_K M_V \leq C\,(M_Q + M_K + M_V)$ and absorbing numeric factors into $c(H, B) = \sqrt{H}\, c_1(B)\, C$ yields the stated form

$$\left\| \text{TST-MHA}(\tilde{z}, \tilde{r}) - \text{TST-MHA}(\tilde{z}', \tilde{r}') \right\|_F \leq c(H, B)\, M_O\,(M_Q + M_K + M_V) \left( \|\tilde{z} - \tilde{z}'\|_F + \|\tilde{r} - \tilde{r}'\|_F \right).$$

$\blacksquare$

## C  ADDITIONAL EXPERIMENTS AND RUNNING TIME

### C.1  ADDITIONAL EXPERIMENTS ON NEW DATASETS

We have also run additional experiments on human mobility (during natural disasters) data (Wang & Taylor, 2016), wildfire data (Mathur et al., 2025), and Twitter data (i.e., geo-tagged Tweets from the United States from January 12 to 18, 2013)). As Table 4 shows, TEN-DM outperforms DSTPP (the next best competitor) on both spatial and temporal dimensions but the gains vary with respect to the dimension and type of the data. In particular, for the spatial dimension on mobility and wildfire datasets, TEN-DM achieves more substantial gains, including significant results on the mobility data ($p$-value $\approx 0.09$; with $^*$), while the difference between TEN-DM and DSTPP on the temporal dimension is negligible. In turn, on the twitter data, TEN-DM outperforms DSTPP with a highly statistically significant gain ($p$-value $\approx 0.001$; with $^{***}$), and yield similar performance on the spatial dimension. These phenomena suggest that TEN-DM captures some inherent latent higher-order structural properties of STTPs that its non-topological competitors such as DSTPP cannot.

Table 4: Performance evaluation for predicting both time and space of the next event on human mobility, wildfire, and Twitter datasets.

| Model | Human Mobility | | Wildfire | | Twitter | |
|---|---|---|---|---|---|---|
| | Spatial ↓ | Temporal ↓ | Spatial ↓ | Temporal ↓ | Spatial ↓ | Temporal ↓ |
| DSTPP | 5.7830±0.3163 | 1.0149±0.0030 | 11.4241±0.0232 | 0.3602±0.0004 | 0.0761±0.0001 | 0.7102±0.0153 |
| **TEN-DM** | **5.5592±0.1387**$^*$ | **1.0130±0.0138** | **11.4084±0.0383** | **0.3600±0.0001** | **0.0760±0.0001** | **0.6774±0.0091**$^{***}$ |

### C.2  ADDITIONAL ABLATION STUDIES

We have conducted an additional ablation study on Theft data (Table 5 below). We observe that removing GNN pre-training/temporal query-guided self-attention mechanism (TQ-SA)/temporal topological learning (TTL) causes performance degradation in both spatial and temporal dimensions. Specifically, on the temporal dimension, compared to without GNN pre-training or TTL, the performance degradation is statistically significant ($^*$).

Table 5: Additional ablation study on Theft data.

| Model | Theft | |
|---|---|---|
| | Spatial ↓ | Temporal ↓ |
| **TEN-DM** | **0.0700±0.0001** | **0.363±0.017** |
| W/o Graph | 0.0701±0.0001 | $^*$0.391±0.015 |
| W/o TQ-SA | 0.0702±0.0001 | 0.374±0.020 |
| W/o TTL | 0.0701±0.0001 | $^*$0.425±0.001 |

To evaluate the sensitivity of topological hyperparameters, including filtration resolution, patch size, and zigzag directionality, we conducted additional ablation experiments using the 311 service dataset. As Table 6 shows, replacing the original zigzag persistence configuration with an alternative filtration resolution (using a smaller step increment), patch size (using the doubled the original patch size), zigzag direction (using the reverse time direction), and weighted multi-scale ZIP fusion yields only marginal performance changes across both temporal and spatial predictions: $0.7920 \pm 0.026$ vs. $0.7928 \pm 0.0091$ or $0.0547 \pm 0.0002$ vs. $0.0549 \pm 0.0002$. These results demonstrate that TEN-DM is robust to variations in topological hyperparameters, and the overall improvements from these modifications are extremely small. This also suggests that our model's performance tends to be **not sensitive** to the specific choices of topological settings, which makes it particularly attractive for limited data availability or sparse data regimes.

We have also conducted additional experiments comparing the TEN-DM model using zigzag persistence versus using standard persistent homology (PH) on the 311 service dataset. We found that

Table 6: Additional ablation study on topological hyperparameters.

| Model | 311 Service | |
|---|---|---|
| | Spatial ↓ | Temporal ↓ |
| TEN-DM (original) | 0.0547±0.0002 | **0.7920±0.0260** |
| TEN-DM with New Filtration Resolution | 0.0548±0.0002 | 0.7928±0.0091 |
| TEN-DM with New Patch Size | 0.0549±0.0002 | 0.8040±0.0072 |
| TEN-DM with Another Zigzag Directionality | **0.0546±0.0002** | 0.7990±0.0750 |
| TEN-DM with Weighted ZPI | 0.0547±0.0001 | 0.8029±0.0014 |

TEN-DM with ZP (i.e., 0.793±0.026 on the temporal dimension) outperforms TEN-DM with ordinary PH (i.e., 0.800±0.008 on the temporal dimension), which can be intuitively expected. Indeed, while both ZP and PH distill inherent higher-order properties of the underlying data generating STPP, ZP captures the most essential topological properties along the temporal dimension.

On the spatial resolution, the optimal spatial resolution can be found by the cross-validation. Tables 7 shows spatio-temporal prediction performance comparison between $256 \times 256$ tiles (i.e., TEN-DM) and $64 \times 64$ tiles (i.e., TEN-DM with new resolution). Overall, we have found that the results are stable with respect to the considered resolutions, which indicates the overall robustness of TEN-DM.

Table 7: Additional ablation study on the spatial resolution.

| Model | 311 Service | |
|---|---|---|
| | Spatial ↓ | Temporal ↓ |
| TEN-DM (original) | **0.0547±0.0002** | **0.7920±0.0260** |
| TEN-DM with New Resolution | 0.0548±0.0002 | 0.8082±0.0087 |

## C.3 SENSITIVE ANALYSIS

In the datasets we consider, we have 0.04% of such events in the 311 Service and 0 such events in JPN Earthquake, US Earthquake, and COVID-19. The model performance for the 311 data with or without the restriction on no duplicate events is essentially the same, while the standard errors become smaller (see Table 8 below).

Table 8: Additional experiments on 311 service data (with and without duplicated events).

| Model | 311 Service | |
|---|---|---|
| | Spatial ↓ | Temporal ↓ |
| TEN-DM w/o Duplicated Events | 0.0547±0.0001 | 0.7920±0.013 |
| **TEN-DM** | 0.0547±0.0002 | 0.7920±0.026 |

To find the threshold $\epsilon$, we use the standard cross-validation argument. In the considered case studies, we have found that our results are overall stable with respect to the selected $\epsilon$ (see Table 9 with $\epsilon$ over the set $[0.1, 0.2, 0.3, 0.4, 0.5]$). However, we agree with the Reviewer that it may be a problem in general. Following the line of research on information criteria, we also suggest to consider a tradeoff between the model performance and network sparsity during the cross-validation, with a penalty associated for the increased computational costs. Alternatively, optimal $\epsilon$ can be selected as Bayesian prior (i.e., "expert knowledge"). Furthermore, the aggregation weights $\alpha_r$ can be configured as fixed hyperparameters or optimized as trainable parameters during model training. The weights $\alpha_r$ will be updated adaptively, when treated as learnable parameters. We have conducted new experiments by adding an adjacency matrix with another threshold $\epsilon$ and set $\alpha_r$ to be trainable. The results on the 311 data (Table 9) indicate that adding an additional adjacency matrix and setting $\alpha_r$ to be trainable can improve prediction performance in the temporal dimension. In the spatial dimension, the updated model's performance is a bit worse than the previous version, however, it is still better than other baselines.

Table 9: Additional experiments on 311 service data with different $\epsilon$.

| Model | 311 Service | |
| --- | --- | --- |
| | Spatial ↓ | Temporal ↓ |
| TEN-DM with adaptive $\alpha_r$ | 0.0551±0.0002 | **0.5978±0.2050** |
| TEN-DM with $\epsilon = 0.1$ | **0.0547±0.0002** | 0.7920±0.0260 |
| TEN-DM with $\epsilon = 0.2$ | 0.0555±0.0002 | 0.8206±0.0223 |
| TEN-DM with $\epsilon = 0.3$ | 0.0550±0.0001 | 0.8176±0.0205 |
| TEN-DM with $\epsilon = 0.4$ | 0.0552±0.0003 | 0.8307±0.0011 |
| TEN-DM with $\epsilon = 0.5$ | 0.0551±0.0003 | 0.8093±0.0025 |

To explore this hypothesis of our TEN-DM, i.e., useful under the conditions of noisy, we randomly select 5% of the latitude and longitude coordinate pairs in the test set of the 311 service data and perturb them by adding Gaussian noise (with $\mu = 0$ and $\sigma = 0.01$). As Table 10 demonstrates, our TEN-DM outperforms DSTPP on both spatial and temporal dimensions. Specifically, on the temporal dimension, TEN-DM achieves statistically significant performance (with $p$-value $\approx 0.02$). In fact, this is not surprising as TEN-DM is equipped with the most essential higher order properties distilled via ZP.

Table 10: Robustness analysis on 311 service data.

| Model | 311 Service | |
| --- | --- | --- |
| | Spatial ↓ | Temporal ↓ |
| DSTPP | 0.8200±0.0034 | 0.0550±0.0002 |
| **TEN-DM** | **0.7879±0.0168** | **0.0548±0.0001** |

## C.4 RUNNING TIME OF ZPI GENERATION

**Remark C.1** (**Running Time of ZPI Generation**). We generated the ZPI representations offline on a local CPU machine (Apple M4 Pro with 24GB memory). For instance, when working on images with a resolution of $20 \times 20$ and sequence length of 1, the average processing time of 0.00040 seconds per event. When the sequence length increases to 5 (with the same resolution), the average processing time of 0.00046 seconds per event. For images with a resolution of $50 \times 50$ and sequence length of 1, the average processing time of 0.0013 seconds per event.

