# OpenReview forum: "TEN-DM: Topology-Enhanced Diffusion Model for Spatio-Temporal Event Prediction"
_ICLR.cc/2026/Conference — ICLR 2026 Poster_

### Official Review · Reviewer_UFd9 · 2025-10-30

**Soundness:** 3
**Presentation:** 2
**Contribution:** 2
**Rating:** 4
**Confidence:** 4

**Summary:**

This paper proposes a TEN-DM framework for modeling Spatio-Temporal Point Processes (STPPs). The method integrates three key components: a Graph Construction and Learning (GCL) module to represent STPP data as graphs, a Temporal Topological Learning (TTL) framework using zigzag persistence to capture evolving topological features, and a Temporal Query-guided Self-Attention (TQ-SA) mechanism. The authors combine these into a diffusion model to learn complex spatio-temporal dependencies, particularly under sparse and noisy regimes. Extensive experiments on five real-world datasets are presented with promising results.

**Strengths:**

Strengths:

- The paper proposes several advanced paradigms, such as the graph neural networks and diffusion models, into a single framework for STPP modeling.

- The paper provides a thorough experimental section, benchmarking TEN-DM against a set of baselines across multiple real-world datasets.

- The proposed model is decomposed into clear, modular components, making the architecture easy to understand.

**Weaknesses:**

Weaknesses:

- One of the main issues of this paper is the lack of motivation and problem analysis. Although there are some discussions for the choices of the key components, the justifications are somewhat insufficient. For example, the rationale for using graph abstraction is "graph abstraction offers a flexible... framework" and "never been used". This does not articulate what specific challenges or limitations exist in current STPP methods.

- Similarly, the motivation for using diffusion models is that they are "a new powerful machinery" and have not been applied to STPPs before, which is somewhat insufficient. Thus, I suggest that to provide a theoretical analysis to show that under what conditions the proposed method is better than the previous methods.

- Furthermore, it would be better if it could provide some case studies to illustrate the key increment and its intuition compared with other STTP methods.

**Questions:**

See the weaknesses above.

---

> ### Author Response · Authors · 2025-11-21
> **Response to Reviewer UFd9 (1/2)**
>
> We thank the reviewer for the detailed and constructive suggestions. Regarding the concerns of the Reviewer UFd9, we provide the following responses.
>
> **Q1:** One of the main issues of this paper is the lack of motivation and problem analysis. Although there are some discussions for the choices of the key components, the justifications are somewhat insufficient. For example, the rationale for using graph abstraction is "graph abstraction offers a flexible... framework" and "never been used". This does not articulate what specific challenges or limitations exist in current STPP methods.
>
> **A:** Thank you for this question, as it’s essential for highlighting the core versatility of DEN. Essentially, most approaches for spatio-temporal processes, including STTPs, rely on the first two moments (i.e., mean and covariance) and Euclidean distance among locations. This poses a number of fundamental limitations in applications. First, in many real world case studies, especially, involving Earth sciences, ecology, and healthcare, the Euclidean distance {\bf may not reflect well} the dependency among the observed events. For example, consider 3 wildfire events at locations A, B, and C. The Euclidean distance between A and B is lower than that of A and C. However, A and B are separated by a mountain and belong to different climatic regimes, while A and C have the same land use and topographic properties. Then it is reasonable to assume that similarity between A and C will be higher than that of A and B. (Alternatively, we can say that this underlying data generating STTP lives on some latent manifold, and Euclidean distance is not an appropriate metric.) Second, spatio-temporal dependency may be both nonlinear, nonstationary and nonseparable, e.g., the intensity function of STTP over space may vary in time, and vice versa (i.e., dependence over spatial locations will change over different seasons, time of the date etc). Third, the dependence structure may also involve higher order terms, requiring the analysis of cumulants, rather than just the first two moments. To address these fundamental challenges, we can leverage a representation in a form of a graph abstraction, where similar (potentially, distant) events are connected via an edge, and we can systematically analyze higher-order dependencies (not just covariances between two locations but interdependencies among multiple interconnected nodes), by leveraging graph theoretic principles and ML for graphs.
>
> While graph abstractions have been used for TPPs, to the best of our knowledge, this is the 1st paper proposing a graph representation for STPPs.

---

> > ### Author Response · Authors · 2025-11-21
> > **Response to Reviewer UFd9 (2/2)**
> >
> > **Q2:** Similarly, the motivation for using diffusion models is that they are "a new powerful machinery" and have not been applied to STPPs before, which is somewhat insufficient. Thus, I suggest that to provide a theoretical analysis to show that under what conditions the proposed method is better than the previous methods. Furthermore, it would be better if it could provide some case studies to illustrate the key increment and its intuition compared with other STTP methods.
> >
> > **A:** Diffusion models, in general, offer higher flexibility for capturing simultaneously both fine-grained and global characteristics of the underlying data generating processes as well as maintaining higher level coherence than other ML and statistical models. This is particularly important for STTPs which  are typically characterized by complex heterogeneity in both space and time,  as the time and space occurrences are highly entangled with each other. We believe that this is the primary reason why just over the last couple of years diffusion models quickly have become a popular alternative for TPP modeling (see some of the the first papers and references therein like Lüdke et al, 2023 and the most recent ones like Zhang et al, 2024, Mukherjee et al., 2025), and we advocate that given the sophisticated interdependencies of STTPs, diffusion models will also become the widely adopted choice for a broad range of STTP learning tasks.
> >
> > Given this premise, we advocate that TEN-DM is particularly useful under the conditions of noisy, sparse or limited data availability.  To explore this hypothesis we randomly select 5% of the latitude and longitude coordinate pairs in the test set of the 311 service data and perturb them by adding Gaussian noise (with $\mu =0$ and $\sigma = 0.01$). As Tables 1 and 2 demonstrate, our TEN-DM outperforms DSTPP on both spatial and temporal dimensions. Specifically, on the temporal dimension, TEN-DM achieves statistically significant performance (with $p$-value $\approx 0.02$; with *). In fact, this is not surprising as TEN-DM is equipped with the most essential higher order properties distilled via ZP. **We have added these results and discussions to the revised version, Appendix C.3 (see Table 10, Page 20)**.
> >
> > Table 1. Temporal prediction performance comparison on perturbed 311 service data (in RMSE)
> > ||TEN-DM | DSTPP |
> > |-|-|-|
> > | 311 | $*$ **0.7879±0.0168** | 0.8200±0.0034 |
> >
> > Table 2. Spatial prediction performance comparison on perturbed 311 service data (in RMSE)
> > ||TEN-DM | DSTPP |
> > |-|-|-|
> > | 311 | **0.0548±0.0001** | 0.0550±0.0002 |
> >
> > We are also running additional experiments with missing data.

---

> > > ### Author Response · Authors · 2025-12-03
> > > **Additional Response to Reviewer UFd9**
> > >
> > > Dear Reviewer UFd9,
> > >
> > > We have conducted an additional experiment with missing data on 311 service data. Tables 3 and 4 report the temporal and spatial prediction performance on 311 service data with missing events. For temporal prediction, TEN-DM achieves an RMSE of 0.7879 ± 0.0168, while DSTPP records 0.7852 ± 0.0453. For spatial prediction, TEN-DM obtains an RMSE of 0.0548 ± 0.0001, compared with 0.0550 ± 0.0002 for DSTPP. Overall, TEN-DM performs better than the baseline by delivering a smaller RMSE in spatial prediction and demonstrating a much smaller standard deviation in temporal prediction, indicating more accurate and more stable performance.
> > >
> > > Table 3. Temporal prediction performance comparison on 311 service data with missing events (in RMSE)
> > > ||TEN-DM | DSTPP |
> > > |-|-|-|
> > > | 311 | 0.7879±0.0168 | 0.7852±0.0453 |
> > >
> > > Table 4. Spatial prediction performance comparison on 311 service data with missing events (in RMSE)
> > > ||TEN-DM | DSTPP |
> > > |-|-|-|
> > > | 311 | 0.0548±0.0001 | 0.0550±0.0002 |
> > >
> > > Best,
> > >
> > > Authors

---

> ### Author Response · Authors · 2025-11-25
> **Any Additional Questions (Reviewer UFd9)?**
>
> Dear Reviewer UFd9,
>
> Thanks very much for providing the constructive and motivating feedback! Can you please let us know whether we have addressed all your questions and whether you have any additional feedback?
>
> Thank you!
>
> Best,
>
> Authors

---

> ### Author Response · Authors · 2025-11-26
> **We are looking forward to hearing from you (Reviewer UFd9)**
>
> Dear Reviewer UFd9,
>
> We hope the updates and explanations resolve the points you raised. If any questions remain, we would like to clarify further. We sincerely appreciate the time you dedicated to reviewing our work.
>
> Thanks very much again!
>
> Best,
>
> Authors

---

### Official Review · Reviewer_Wz4A · 2025-10-31

**Soundness:** 2
**Presentation:** 2
**Contribution:** 2
**Rating:** 2
**Confidence:** 4

**Summary:**

This paper presents TEN-DM, a Topology-Enhanced Diffusion Model for spatio-temporal point process (STPP) prediction. The authors aim to address the limitations of existing deep learning models in capturing complex, non-stationary spatio-temporal dependencies, especially under sparse and noisy conditions. TEN-DM introduces three core components: (i) a graph construction and learning module to model event interactions, (ii) a temporal topological learning (TTL) framework based on zigzag persistence to extract dynamic topological features from time-series images, and (iii) a temporal query-guided self-attention (TQ-SA) mechanism to capture periodic patterns. The model is evaluated on five real-world datasets and outperforms the baselines in both spatial and temporal prediction tasks.

**Strengths:**

- Originality: This is the first work to integrate zigzag persistence and diffusion models for STPP forecasting. The use of topological data analysis (TDA) in the form of zigzag persistence images (ZPI) to capture time-evolving shape patterns is novel and well-motivated. The idea of converting STPP data into image time-series and analyzing their topological evolution is creative and technically sound.

- Technical Quality: The paper is mathematically rigorous, with formal definitions of cubical complexes, filtrations, and zigzag persistence. The stability theorem (Theorem 3.2) provides theoretical grounding for the robustness of ZPI under noise. The Lipschitz bound for the proposed attention mechanism (TST-MHA) further demonstrates the model’s theoretical controllability.

- Clarity: Despite the complexity of the methodology, the paper is well-organized and clearly written. Each module is introduced with both intuition and formalization. Figures (e.g., Fig. 1, Fig. 2) effectively illustrate the pipeline and help readers understand the workflow.

- Significance: The paper addresses a real-world problem, forecasting discrete events in space and time (e.g., earthquakes, crimes, disease outbreaks), and proposes a unified, interpretable, and robust solution. The integration of geometry, topology, and diffusion opens a new research direction in spatio-temporal modeling, especially for sparse and irregular data.

**Weaknesses:**

- Scalability and Efficiency Concerns: While the model is effective on small-scale datasets (e.g., ~10K events), its scalability to large-scale urban data (e.g., millions of taxi trips or tweets) is unclear. The zigzag persistence computation and image rasterization steps may become prohibitively expensive for high-resolution or long-duration data. A complexity breakdown or runtime scaling analysis is missing.

- Limited Ablation on Topological Hyperparameters: The paper does not thoroughly explore the sensitivity of ZPI parameters, such as filtration resolution, patch size, or zigzag directionality. The multi-scale mixing uses fixed weights (βq = 0.25), but adaptive or learned weighting could be more effective. An ablation on these choices would strengthen the contribution.

- Generalization Across Domains: Although the model is tested on five datasets, they are all from the US or Japan, and mostly urban or seismic events. There is no evaluation on human mobility, social media, or climate events, which are also common STPP scenarios. A cross-domain generalization test would better support the claim of universality.

- Baseline Diversity: While 17 baselines are included, few recent graph-based or transformer-based STPP models are missing. Also, no comparison with other TDA-based methods is provided. This limits the completeness of the empirical evaluation.

**Questions:**

- Scalability: How does TEN-DM scale to larger datasets (e.g., >1M events)? What is the time and memory complexity of ZPI generation and TTL module with respect to image resolution and sequence length?

- Topological Sensitivity: How does the model performance change with different filtration functions, patch sizes, or zigzag directions? Have you tried adaptive weighting for multi-scale ZPI fusion?

- Cross-domain Evaluation: Have you tested TEN-DM on non-urban, non-seismic data, such as animal movement, social media check-ins, or climate anomalies? How general is the topological assumption?

- Comparison with TDA Baselines: Why not compare with other TDA-enhanced models? This would better highlight the unique value of zigzag persistence.

- Real-time Forecasting: Is TEN-DM suitable for real-time or online forecasting? Can the ZPI and TTL modules be incrementally updated as new events arrive?

---

> ### Author Response · Authors · 2025-11-21
> **Response to Reviewer Wz4A (1/3)**
>
> We thank the reviewer for the detailed and constructive suggestions. Regarding the concerns of the Reviewer Wz4A, we provide the following responses.
>
> **Q1:** Limited Ablation on Topological Hyperparameters: The paper does not thoroughly explore the sensitivity of ZPI parameters, such as filtration resolution, patch size, or zigzag directionality. The multi-scale mixing uses fixed weights (\beta_p = 0.25), but adaptive or learned weighting could be more effective. An ablation on these choices would strengthen the contribution. Topological Sensitivity: How does the model performance change with different filtration functions, patch sizes, or zigzag directions? Have you tried adaptive weighting for multi-scale ZPI fusion?
>
> **A:** Thanks very much for this question as it inherently relates to the robustness of TEN which is a particularly important advantage under limited or sparse data scenarios.
>
> To evaluate the sensitivity of topological hyperparameters, including filtration resolution, patch size, and zigzag directionality, we conducted additional ablation experiments using the 311 service dataset. As Tables 1 and 2 show, replacing the original zigzag persistence configuration with an alternative filtration resolution (using a smaller step increment), patch size (using the doubled the original patch size), zigzag direction (using the reverse time direction), and weighted multi-scale ZIP fusion yields only marginal performance changes across both temporal and spatial predictions:
> 0.7920±0.026 vs. 0.7928±0.0091 or 0.0547±0.0002 vs. 0.0549±0.0002.
>
> These results demonstrate that TEN-DM is robust to variations in topological hyperparameters, and the overall improvements from these modifications are extremely small. This also suggests that our model’s performance tends to be {\bf not sensitive} to the specific choices of topological settings, which makes it particularly attractive for limited data availability or sparse data regimes.
>
> Table 1. Temporal prediction performance comparison on 311 data
> |TEN-DM | TEN-DM with new filtration resolution | TEN-DM with new patch size | TEN-DM with another zigzag directionality | TEN-DM with weighted ZPI (different patch size) |
> |-|-|-|-|-|
> | 311 | **0.7920±0.0260 | 0.7928±0.0091 | 0.8040±0.0072 | 0.7990±0.0750 | 0.8029±0.0014|
>
>
> Table 2. Spatial prediction performance comparison on 311 data
> |TEN-DM | TEN-DM with new filtration resolution | TEN-DM with new patch size | TEN-DM with another zigzag directionality | TEN-DM with weighted ZPI (different patch size) |
> |-|-|-|-|-|
> | 311 | 0.0547±0.0002 | 0.0548±0.0002 | 0.0549 ±0.0002 | **0.0546±0.0002** | 0.0547±0.0001
>
> We are currently conducting additional sensitivity analyses on other datasets and will report the results once they are completed.
>
> **We have added the above results and discussions to the revised version, Appendix C.2 (see Table 6, Page 19)**.

---

> ### Author Response · Authors · 2025-11-21
> **Response to Reviewer Wz4A (2/3)**
>
> **Q2:** Generalization Across Domains: Although the model is tested on five datasets, they are all from the US or Japan, and mostly urban or seismic events. There is no evaluation on human mobility, social media, or climate events, which are also common STPP scenarios. A cross-domain generalization test would better support the claim of universality. Cross-domain Evaluation: Have you tested TEN-DM on non-urban, non-seismic data, such as animal movement, social media check-ins, or climate anomalies? How general is the topological assumption?
>
> **A:** Thanks very much for this question that allows us to illustrate the generalizability of TEN-DM! We have run additional experiments on human mobility (during natural disasters) data [1], wildfire data [2], and Twitter data (i.e., geo-tagged Tweets from the United States from January 12 to 18, 2013)​), covering additional domains such as human mobility, social media, or climate events. As Tables 3-8 demonstrate, TEN-DM outperforms DSTPP (the next best competitor) on both spatial and temporal dimensions but the gains vary with respect to the dimension and type of the data. In particular, for the spatial dimension on mobility and wildfire datasets, TEN-DM achieves more substantial gains, including significant results on the mobility data  ($p$-value $\approx 0.09$; with *), while the difference between TEN-DM and DSTPP on the temporal dimension is negligible. In turn, on the twitter data, TEN-DM outperforms DSTPP with a highly statistically significant gain ($p$-value $\approx 0.001$; with ***), and yield similar performance on the spatial dimension. These phenomena suggest that TEN-DM captures some inherent latent higher-order structural properties of STTPs that its non-topological competitors such as DSTPP cannot.
>
> If the Reviewer finds it interesting, we can run additional experiments on the healthcare and biosurveillance datasets.
>
> Table 3. Temporal prediction performance comparison on human mobility data (in RMSE)
> ||TEN-DM | DSTPP |
> |-|-|-|
> | Human mobility | **1.0130±0.0138** | 1.0149±0.0030
>
> Table 4. Spatial prediction performance comparison on human mobility data (in RMSE)
> ||TEN-DM | DSTPP |
> |-|-|-|
> | Human mobility | $^*$ **5.5592±0.1387** | 5.7830±0.3163
>
>
> Table 5. Temporal prediction performance comparison on wildfire data (in RMSE)
> ||TEN-DM | DSTPP |
> |-|-|-|
> | Wildfire | **0.3600±0.0001** | 0.3602±0.0004 |
>
> Table 6. Spatial prediction performance comparison on wildfire data (in RMSE)
> ||TEN-DM | DSTPP |
> |-|-|-|
> | Wildfire | **11.4084±0.0383** | 11.4241±0.0232 |
>
> Table 7. Temporal prediction performance comparison on twitter data (in RMSE)
> ||TEN-DM | DSTPP |
> |-|-|-|
> | Twitter | $^{***}$ **0.6774±0.0091** | 0.7102±0.0153 |
>
> Table 8. Spatial prediction performance comparison on twitter data (in RMSE)
> ||TEN-DM | DSTPP |
> |-|-|-|
> | Twitter | **0.0760±0.0001** | 0.0761±0.0001 |
>
> [1] Wang Q, Taylor JE (2016) Patterns and limitations of urban human mobility resilience under the influence of multiple types of natural disaster. PLoS ONE 11(1): e0147299.
>
> [2] Spatiotemporal Wildfire Prediction and Reinforcement Learning for Helitack Suppression, ICMLA 2025.
>
> **We have added these results to the revised version, Appendix C.1 (see Table 4, Page 18)**.

---

> > ### Author Response · Authors · 2025-11-21
> > **Response to Reviewer Wz4A (3/3)**
> >
> > **Q3:** Baseline Diversity: While 17 baselines are included, few recent graph-based or transformer-based STPP models are missing. Also, no comparison with other TDA-based methods is provided. This limits the completeness of the empirical evaluation.
> >
> > **A:** We thank the reviewer for the valuable feedback. As suggested by the Reviewer, we have added two most recent graph-based or transformer-based STPP models, particularly, a graph-based TPP model [1] (i.e., GraDK which is the deep graph kernel) and one transformer-based STPP model [2]. Table 5 shows the comparative results on COVID-19 data (we selected this dataset as [2] considered it too, but we are running additional experiments on other datasets). As Table 5 shows, our TEN-DM achieves {\bf substantially} better temporal-dimension prediction performance compared to both graph- and transformer-based models.
> >
> > Table 5. Temporal prediction performance comparison on COVID-19 (in RMSE).
> > ||TEN-DM | GraDK | EventFormer |
> > |-|-|-|-|
> > | COVID-19 | **0.087** | 0.098 | 0.100
> >
> > To the best of our knowledge, there is no TDA-based STPP model. In fact, we believe that TEN-DM is the 1st paper introducing PH and TDA, in general, to ML for STPP.
> >
> > However, we have conducted additional experiments comparing the TEN-DM model using zigzag persistence versus using standard persistent homology (PH) on the 311 service dataset. As Table 6 demonstrates, TEN-DM with ZP outperforms TEN-DM with ordinary PH, which can be intuitively expected. Indeed, while both ZP and PH distill inherent higher-order properties of the underlying data generating STPP, ZP captures the most essential topological properties along the temporal dimension. **We have added the above results and discussions to the revised version, Appendix C.2 (see the text highlighted in blue (lines #971-985), Page 19)**.
> >
> > Table 6. Temporal prediction performance comparison on 311 service data (in RMSE).
> > ||TEN-DM | TEN-DM with regular PH |
> > |-|-|-|
> > | 311 | **0.793±0.026** |  0.800±0.008
> >
> > [1] Dong, Z., Mateu, J., and Xie, Y.. Spatio-temporal network point processes for modeling crime events with landmarks. (2025) https://arxiv.org/html/2409.10882v2
> >
> > [2] Yu, S., Guo, D., Fu, Y. and Jin, W. (2025). EventFormer: a hierarchical neural point process framework for spatio-temporal clustering events prediction. Journal of Big Data, 12(1), p.162.
> >
> > **Q4:** Scalability: How does TEN-DM scale to larger datasets (e.g., >1M events)? What is the time and memory complexity of ZPI generation and TTL module with respect to image resolution and sequence length?
> >
> > **A:** We thank the reviewer for the valuable feedback. We generated the ZPI representations offline on a local CPU machine (Apple M4 Pro with 24GB memory). For images with a resolution of $20 \times 20$ and sequence length of 1, the average processing time of 0.00040 seconds per event. Thus, generating ZPI for 1 million events takes approximately 7 minutes. For images with a resolution of $20 \times 20$ and sequence length of 5, the average processing time of 0.00046 seconds per event. Thus, generating ZPI for 1 million events takes approximately 8 minutes.
> >
> > For images with a resolution of $50 \times 50$ and sequence length of 1, the average processing time of 0.0013 seconds per event. Thus, generating ZPI for 1 million events takes approximately 24 minutes. For images with a resolution of $50 \times 50$ and sequence length of 5, the average processing time of 0.0014 seconds per event. Thus, generating ZPI for 1 million events takes approximately 25 minutes.
> >
> > Hence, overall, ZPIs for cubical complexes appear noticeably less computationally expensive than PH for other types of the data representations. **We have added a remark of running time of ZPI generation to the revised version, Appendix C.4 (Page 20)**.
> >
> > **Q5:** Real-time Forecasting: Is TEN-DM suitable for real-time or online forecasting? Can the ZPI and TTL modules be incrementally updated as new events arrive?
> >
> > **A:** Thanks very much for suggesting this interesting research direction! To the best of our knowledge, neither ZP nor even PH has ever been incrementally updated in a sequential regime. However, there are some recent attempts to adaptively approximate PH with neural networks (Nishikawa et al., NeurIPS'23). We believe that this idea can be potentially extended for incremental adaptation in a sequential regime. However, this is a fundamental open problem, both in ML and computational topology, which we leave as future research direction.
> >
> > On the TTL module in the sequential regime, it can be incrementally updated through the `pre-training and fine-tune’ paradigm. That is, we can integrate historical temporal topological features, while preserving the universal knowledge during the pre-training phase. Then, we can adopt parameter efficient fine-tuning (PEFT) strategies to update a small portion of the parameters and freeze the remaining parameters, which also helps mitigate the risk of overfitting.

---

> > > ### Author Response · Authors · 2025-11-25
> > > **Additional Response to Reviewer Wz4A**
> > >
> > > Dear Reviewer Wz4A,
> > >
> > > We have completed additional experiments on Japan Earthquake data. As shown in the below Tables 1 and 2, we observe that TEN-DM with alternative topological hyperparameter configurations achieve better performance on the temporal dimension. Meanwhile, the performance on the spatial dimension remains stable across different setups.
> > >
> > > Table 1. Temporal prediction performance comparison on Japan Earthquake data.
> > > |TEN-DM | TEN-DM with new filtration resolution | TEN-DM with new patch size | TEN-DM with another zigzag directionality | TEN-DM with weighted ZPI (different patch size) |
> > > |-|-|-|-|-|
> > > | Japan Earthquake | 0.3710±0.0030 | 0.3634±0.0025 | 0.3647±0.0039 | 0.3654±0.0036 | 0.3626±0.0027 |
> > >
> > >
> > > Table 2. Spatial prediction performance comparison on Japan Earthquake data.
> > > |TEN-DM | TEN-DM with new filtration resolution | TEN-DM with new patch size | TEN-DM with another zigzag directionality | TEN-DM with weighted ZPI (different patch size) |
> > > |-|-|-|-|-|
> > > | Japan Earthquake | 6.6470±0.0410 | 6.6731±0.0489 | 6.5898±0.0446 | 6.6505±0.0081 | 6.5619±0.0614 |
> > >
> > > Best,
> > >
> > > Authors

---

> ### Author Response · Authors · 2025-11-25
> **Any Additional Questions (Reviewer Wz4A)?**
>
> Dear Reviewer Wz4A,
>
> Thanks very much for providing the constructive and motivating feedback! Can you please let us know whether we have addressed all your questions and whether you have any additional feedback?
>
> Thank you!
>
> Best,
>
> Authors

---

> ### Author Response · Authors · 2025-11-26
> **We are looking forward to hearing from you (Reviewer Wz4A)**
>
> Dear Reviewer Wz4A,
>
> We hope the updates and explanations resolve the points you raised. If any questions remain, we would like to clarify further. We sincerely appreciate the time you dedicated to reviewing our work.
>
> Thanks very much again!
>
> Best,
>
> Authors

---

> > ### Comment · Reviewer_Wz4A · 2025-11-27
> > **Thanks for rebuttal.**
> >
> > Thank you for the authors' reply. As my concerns have largely been clarified, I have updated the evaluation.
> >
> > Best,
> >
> > Reviewer Wz4A

---

> > > ### Author Response · Authors · 2025-11-27
> > > **Thanks very much, Reviewer Wz4A!**
> > >
> > > Dear Reviewer Wz4A,
> > >
> > > Thanks very much for the constructive feedback and, of course, for raising the score! We will add all suggested edits to the final version. We sincerely appreciate your thoughtful comments and valuable input that helped strengthen our paper.
> > >
> > > Best,
> > >
> > > Authors

---

> ### Author Response · Authors · 2025-11-27
> **A one more comment**
>
> Dear Reviewer Wz4A,
>
> Just one more comment, if you think that it's appropriate, we'd be very grateful if you consider championing for the paper if possible. We have a few more experiments that we hope to complete by the end of the rebuttal period to make the case even stronger.
>
> Thank you!
>
> Authors.

---

### Official Review · Reviewer_rwpn · 2025-11-01

**Soundness:** 2
**Presentation:** 3
**Contribution:** 2
**Rating:** 6
**Confidence:** 3

**Summary:**

The paper introduces Topology-ENhanced Diffusion Model (TEN-DM) for predicting future events in spatio-temporal point process (STPP) data.

The central problem the authors address is that existing methods often fail to capture the complex, higher-order dependencies between the spatial and temporal dimensions. Their solution is a conditional diffusion model where the conditioning signal is a sophisticated, fused representation of the event history. It fuses Spatio-Temporal Graph, Spatial information, temporal information and Topological Learning (converts the STPP data into a time-series of 2D images). This fused embedding guides the diffusion model’s denoising process to accurately predict both the time and location of the next event.

Experiments on five real-world datasets (e.g., earthquakes, crime, COVID-19) show that TEN-DM achieves state-of-the-art performance

**Strengths:**

The motivation is straightforward: conditioning a spatio-temporal generative model on more available information generally improves performance. The approach proves highly effective, achieving top results across five diverse real-world datasets and outperforming 17 baselines.

**Weaknesses:**

The proposed pipeline is exceptionally complex. It involves GNN pre-training, data-to-image conversion, multi-scale cubical zigzag persistence computation (which is notoriously expensive), a CNN on persistence images, and a conditional diffusion model. This complexity may make the model impractical.

The conversion of event data to a 2D image (Section 3.2) is a critical step, but it is underspecified and potentially lossy. The paper states, "we rasterize the events geo-coordinates onto the 2D image by recording as each pixel's value the associated temporal attribute." But What happens if two or more events fall into the same grid and same time patch? How to choose the grid resolution?

The GCL module feels less developed than the TTL module. The graph is described as a similarity-based $\epsilon$-graph (using cosine similarity), which is known to be highly sensitive to the choice of the threshold $\epsilon$ (or $\mathbb{R}^r$ in the paper). The mechanism by which the aggregation weights $\{\alpha_r\}$ are "updated adaptively" is not explained

**Questions:**

Could the author please clarify the rasterization process in Section 3.2?

Why did you choose the image-based cubical complex representation?

In Section 3.1, how are the adjacency matrix aggregation weights $\{\alpha_r\}$ "updated adaptively"? Are these learnable parameters optimized end-to-end, or set via a separate process?

---

> ### Author Response · Authors · 2025-11-21
> **Response to Reviewer rwpn (1/4)**
>
> We thank the reviewer for the detailed and constructive suggestions. Regarding the concerns of the Reviewer rwpn, we provide the following responses.
>
> **Q1:** The proposed pipeline is exceptionally complex. It involves GNN pre-training, data-to-image conversion, multi-scale cubical zigzag persistence computation (which is notoriously expensive), a CNN on persistence images, and a conditional diffusion model. This complexity may make the model impractical.
>
> **A:** Thank you very much for your thorough and thoughtful comment. We recognize the computational complexity can be a concern.
>
> However, in fact, the multi-scale cubical zigzag persistence computation is very efficient, relative to other topological methods. We generated the ZPI representations offline on a local CPU machine (Apple M4 Pro with 24GB memory). For instance, when working on images with a resolution of $20 \times 20$ and sequence length of 1, the average processing time of 0.00040 seconds per event. When the sequence length increases to 5 (with the same resolution), the average processing time of 0.00046 seconds per event. For images with a resolution of $50 \times 50$ and sequence length of 1, the average processing time of 0.0013 seconds per event. For images with a resolution of $50 \times 50$ and sequence length of 5, the average processing time of 0.0014 seconds per event. **We have added a remark of running time of ZPI generation to the revised version, Appendix C.4 (Page 20)**.
>
> Our ablation study (see Table 2, Section A in the Appendix) shows how these components affect the model performance. In addition, we have conducted a new ablation study on Theft data (Table 1 below). We observe that removing GNN pre-training/temporal query-guided self-attention mechanism (TQ-SA)/temporal topological learning (TTL) causes performance degradation in both spatial and temporal dimensions. Specifically, on the temporal dimension, compared to without GNN pre-training or TTL, the performance degradation is statistically significant ($^*$). **We have added these results and discussions to the revised version, Appendix C.2 (see Table 5, Page 18)**.
>
> Table 1. Temporal prediction performance comparison on Theft data (in RMSE).
> | | Theft (Spatial) | Theft (Temporal) |
> |-|-|-|
> | TEN-DM| 0.0700±0.0001 | 0.363±0.017
> | W/o Graph | 0.0701±0.0001| $^{*}$0.391±0.015
> | W/o TQ-SA | 0.0702±0.0001 | 0.374±0.020
> | W/o TTL | 0.0701±0.0001 | $^{*}$0.425±0.001
>
> We have also run additional experiments on human mobility (during natural disasters) data [1], wildfire data [2], and Twitter data (i.e., geo-tagged Tweets from the United States from January 12 to 18, 2013)​). As Tables 2-7 show, TEN-DM outperforms DSTPP (the next best competitor) on both spatial and temporal dimensions but the gains vary with respect to the dimension and type of the data. In particular, for the spatial dimension on mobility and wildfire datasets, TEN-DM achieves more substantial gains, including significant results on the mobility data  ($p$-value $\approx 0.09$; with *),
> while the difference between TEN-DM and DSTPP on the temporal dimension is negligible. In turn, on the twitter data, TEN-DM outperforms DSTPP with a highly statistically significant gain ($p$-value $\approx 0.001$; with ***), and yield similar performance on the spatial dimension. These phenomena suggest that TEN-DM captures some inherent latent higher-order structural properties of STTPs that its non-topological competitors such as DSTPP cannot. **We have added these results and discussions to the revised version, Appendix C.1 (see Table 4, Page 18)**.
>
> Table 2. Temporal prediction performance comparison on human mobility data (in RMSE)
> ||TEN-DM | DSTPP |
> |-|-|-|
> | Human mobility | **1.0130±0.0138** | 1.0149±0.0030
>
> Table 3. Spatial prediction performance comparison on human mobility data (in RMSE)
> ||TEN-DM | DSTPP |
> |-|-|-|
> | Human mobility | $^*$ **5.5592±0.1387** | 5.7830±0.3163
>
>
> Table 4. Temporal prediction performance comparison on wildfire data (in RMSE)
> ||TEN-DM | DSTPP |
> |-|-|-|
> | Wildfire | **0.3600±0.0001** | 0.3602±0.0004 |
>
> Table 5. Spatial prediction performance comparison on wildfire data (in RMSE)
> ||TEN-DM | DSTPP |
> |-|-|-|
> | Wildfire | **11.4084±0.0383** | 11.4241±0.0232 |
>
> Table 6. Temporal prediction performance comparison on twitter data (in RMSE)
> ||TEN-DM | DSTPP |
> |-|-|-|
> | Twitter | $^{***}$ **0.6774±0.0091** | 0.7102±0.0153 |
>
> Table 7. Spatial prediction performance comparison on twitter data (in RMSE)
> ||TEN-DM | DSTPP |
> |-|-|-|
> | Twitter | **0.0760±0.0001** | 0.0761±0.0001 |
>
> [1] Wang Q, Taylor JE (2016) Patterns and limitations of urban human mobility resilience under the influence of multiple types of natural disaster. PLoS ONE 11(1): e0147299.
>
> [2] Spatiotemporal Wildfire Prediction and Reinforcement Learning for Helitack Suppression, ICMLA 2025.

---

> ### Author Response · Authors · 2025-11-21
> **Response to Reviewer rwpn (2/4)**
>
> **Q2:** The conversion of event data to a 2D image (Section 3.2) is a critical step, but it is underspecified and potentially lossy. The paper states, "we rasterize the events geo-coordinates onto the 2D image by recording as each pixel's value the associated temporal attribute." But What happens if two or more events fall into the same grid and same time patch? How to choose the grid.
>
> **A:** Thank you for pointing it out! Yes, we agree that potentially there may be a situation that two or more events fall into the same grid and same time patch, which will result in the information loss. This situation can be mitigated by imposing the following constraints:
> 1) change spatial resolution so that no grid contains two or more overlapping events;
> 2) if 1) is not possible, split the time patch into two non-overlapping patches, so the condition of no two or more events in the same grid and same time patch is satisfied.
>
> If the fraction of events falling into the same grid and same time patch is relatively low, it negligibly impacts the model performance.  For example, in the datasets we consider, we have  0.04% of such events in the 311 Service and 0 such events in JPN Earthquake, US Earthquake, and COVID-19. The model performance for the 311 data with or without the restriction on no duplicate events is essentially the same, while the standard errors become smaller (see Tables 8 and 9 below). **We have added these results and discussions to the revised version, Appendix C.3 (see Table 8, Page 19)**.
>
> Table 8. Temporal prediction performance comparison on 311 service data (in RMSE)
> |TEN-DM | TEN-DM w/o duplicated |
> |-|-|
> | **0.7920±0.026** | 0.7920±0.0130 |
>
> Table 9. Spatial prediction performance comparison on 311 service data (in RMSE)
> |TEN-DM | TEN-DM w/o duplicated |
> |-|-|
> | **0.0547±0.0002** | 0.0547±0.0001 |
>
> On the spatial resolution, the optimal spatial resolution can be found by the cross-validation. Tables 10 and 11 show spatio-temporal prediction performance comparison between $256 \times 256$ tiles (i.e., TEN-DM) and $64 \times 64$ tiles (i.e., TEN-DM with new resolution). Overall, we have found that the results are stable with respect to the considered resolutions, which indicates the overall robustness of TEN-DM. **We have added these results and discussions to the revised version, Appendix C.2 (see Table 7, Page 19)**.
>
> Table 10. Temporal prediction performance comparison on 311 service data (in RMSE)
> |TEN-DM | TEN-DM with new resolution |
> |-|-|
> | **0.7920±0.026** | 0.8082±0.0087 |
>
> Table 11. Spatial prediction performance comparison on 311 service data (in RMSE)
> |TEN-DM | TEN-DM with new resolution |
> |-|-|
> | **0.0547±0.0002** | 0.0548±0.0002 |

---

> ### Author Response · Authors · 2025-11-21
> **Response to Reviewer rwpn (3/4)**
>
> **Q3:** The GCL module feels less developed than the TTL module. The graph is described as a similarity-based \epsilon-graph (using cosine similarity), which is known to be highly sensitive to the choice of the threshold \epsilon. The mechanism by which the aggregation weights \alpha_r are "updated adaptively" is not explained. In Section 3.1, how are the adjacency matrix aggregation weights \alpha_r "updated adaptively"? Are these learnable parameters optimized end-to-end, or set via a separate process?
>
> **A:** We appreciate the reviewer's constructive comments. To find the threshold $\epsilon$, we use the standard cross-validation argument. In the considered case studies, we have found that our results are overall stable with respect to the selected \epsilon (see Tables 12 and 13 with $\epsilon$ over the set [0.1, 0.2, 0.3, 0.4, 0.5]). However, we agree with the Reviewer that it may be a problem in general. Following the line of research on information criteria, we also suggest to consider a tradeoff between the model performance and network sparsity during the crossvalidation, with a penalty associated for the increased computational costs. Alternatively, optimal $\epsilon$ can be selected as Bayesian prior (i.e., “expert knowledge”).
>
> Table 12. Temporal prediction performance comparison on 311 data under different $\epsilon$
> ||TEN-DM with $\epsilon = 0.1$ | TEN-DM with $\epsilon = 0.2$ | TEN-DM with $\epsilon = 0.3$ | TEN-DM with $\epsilon = 0.4$ | TEN-DM with $\epsilon = 0.5$ |
> |-|-|-|-|-|-|
> | 311 | **0.7925±0.0265** | 0.8206±0.0223 | 0.8176±0.0205 | 0.8307±0.0011 | 0.8093±0.0025 |
>
> Table 13. Spatial prediction performance comparison on 311 data under different $\epsilon$
> ||TEN-DM with $\epsilon = 0.1$ | TEN-DM with $\epsilon = 0.2$ | TEN-DM with $\epsilon = 0.3$ | TEN-DM with $\epsilon = 0.4$ | TEN-DM with $\epsilon = 0.5$ |
> |-|-|-|-|-|-|
> | 311 | **0.0547±0.0002** | 0.0555±0.0002 |0.0550±0.0001 | 0.0552±0.0003 | 0.0551±0.0003  |
>
> The aggregation weights $\alpha_r$ can be configured as fixed hyperparameters or optimized as trainable parameters during model training. The weights $\alpha_r$ will be updated adaptively, when treated as learnable parameters. We have conducted new experiments by adding an adjacency matrix with another threshold $\epsilon$ and set $\alpha_r$ to be trainable. The results on the 311 data (Tables 14 and 15 for temporal and spatial dimensions respectively) indicate  that adding an additional adjacency matrix and setting $\alpha_r$ to be trainable can improve prediction performance in the temporal dimension. In the spatial dimension, the updated model’s performance is a bit worse than the previous version, however, it is still better than other baselines.
>
> Table 14. Temporal prediction performance comparison on 311 data
> ||TEN-DM | TEN-DM with adaptive $\alpha_r$ |
> |-|-|-|
> | 311 | **0.7925±0.0265** | 0.5978±0.2050
>
> Table 15. Spatial prediction performance comparison on 311 data
> ||TEN-DM | TEN-DM with adaptive $\alpha_r$ |
> |-|-|-|
> | 311 | **0.0547±0.0002** | 0.0551±0.0002
>
> **We have added the above results and discussions to the revised version, Appendix C.3 (see Table 9, Page 20)**.

---

> ### Author Response · Authors · 2025-11-21
> **Response to Reviewer rwpn (4/4)**
>
> **Q4:** Could the author please clarify the rasterization process in Section 3.2?
>
> **A:** Thank you for the suggestion. This process is described as follows: ``Within the patch $i$, we rasterize the events’ geo-coordinates onto the 2D image by recording as each pixel’s value the associated temporal attribute’’. Specifically, we convert the spatial locations of events into a grayscale image, where each pixel corresponds to a discretized spatial cell. Pixels containing at least one event are assigned a value of 1 (white), while pixels without events are assigned 0 (black). Thus, we can create a binary image that encodes the spatial distribution of events within the patch. This binary image is then used as the input to construct the filtration for zigzag persistence. **We have added this clarification to the revised version, Subsection 3.2 (see the text highlighted in blue, Page 5)**.
>
>
> **Q5:** Why did you choose the image-based cubical complex representation?
>
> **A:** Thank you for bringing this up. The cubical complex is a standard approach in image analysis, because it is a natural and direct way to describe topological properties of grid-based data, where a grid is viewed as a cell complex with cells of all dimensions (see, e.g., overviews by Kaczynski et al. 2004, Kovalevsky, 2011 and Wagner et al. 2012). As a result, cubical complexes allow us to directly extract topological features of the grid-based data in a computationally and memory efficient manner without triangulation.
>
> Kaczynski, K., Mischaikow, K. and M., Mrozek (2004). Computational Homology. Applied Mathematical Sciences.
>
> Kovalevsky, V. (2011). Introduction to Digital Topology
>
> Wagner, H., Chen, C.,  and E. Vucini (2012). Efficient Computation of Persistent Homology for Cubical Data. Mathematics and Visualization.

---

> ### Author Response · Authors · 2025-11-25
> **Any Additional Questions (Reviewer rwpn)?**
>
> Dear Reviewer rwpn,
>
> Thanks very much for providing the constructive and motivating feedback! Can you please let us know whether we have addressed all your questions and whether you have any additional feedback?
>
> Thank you!
>
> Best,
>
> Authors

---

> > ### Author Response · Authors · 2025-11-26
> > **We are looking forward to hearing from you (Reviewer rwpn)**
> >
> > Dear Reviewer rwpn,
> >
> > We hope the updates and explanations resolve the points you raised. If any questions remain, we would like to clarify further. We sincerely appreciate the time you dedicated to reviewing our work.
> >
> > Thanks very much again!
> >
> > Best,
> >
> > Authors

---

### Author Response · Authors · 2025-12-03
**General Response and Summary**

Dear AC, SAC, PC,

We want to thank all reviewers again for their insightful comments and suggestions which enabled us to substantially improve the paper during the rebuttal phase. We have revised the paper to address the reviewers’ concerns. In the post, we summarize the global context and major revisions for further discussion:

**Positive things**:

**Model design and proposed methods**
- Reviewer Wz4A: ``This is the first work to integrate zigzag persistence and diffusion models for STPP forecasting’’ (Reviewer Wz4A has raised the score from 2 to 6).

- Reviewer Wz4A: ``The paper is mathematically rigorous, with formal definitions of cubical complexes, filtrations, and zigzag persistence.’’ (Reviewer Wz4A has raised the score from 2 to 6).

- Reviewer UFd9: ``The paper proposes several advanced paradigms’’.

- Reviewer UFd9: ``The proposed model is decomposed into clear, modular components, making the architecture easy to understand.’’.

**Empirical evaluation**

- Reviewer rwpn: ``The approach proves highly effective, achieving top results across five diverse real-world datasets and outperforming 17 baselines.’’.

- Reviewer Wz4A: ``The paper addresses a real-world problem, forecasting discrete events in space and time (e.g., earthquakes, crimes, disease outbreaks), and proposes a unified, interpretable, and robust solution.’’ (Reviewer Wz4A has raised the score from 2 to 6).

- Reviewer UFd9: ``The paper provides a thorough experimental section, benchmarking TEN-DM against a set of baselines across multiple real-world datasets.’’.

**Rating score revision**: We got the reply from Reviewer Wz4A - ``As my concerns have largely been clarified, I have updated the evaluation.’’ and **Reviewer Wz4A has raised the score from 2 to 6**.

**Major revisions**:
1. We have added experiments on 3 large-scale and cross-domain datasets (i.e., human mobility, wildfire, and Twitter) and shown our TEN-DM consistently achieves best performance across both spatial and temporal dimensions compared with the runner-up (Reviewer Wz4A).

2. We have incorporated state-of-the-art baselines for comparison and demonstrated that TEN-DM outperforms all of them (Reviewer Wz4A).

3. We have added additional ablation studies (extra four alternative topological hyperparameter configurations on two datasets) on topological hyperparameters and provided discussions on the selection of topological hyperparameters (Reviewer Wz4A).

4. We have conducted additional ablation studies on the Theft data and shown the importance of the different components within TEN-DM (Reviewer rwpn). Furthermore, we provided running time details and clarified the efficiency of multi-scale cubical zigzag persistence computation (Reviewers rwpn \& Wz4A).

5. We have explored two situations, i.e., (a) two or more events fall into the same grid and same time patch and (b) the optimal spatial resolution for multi-scale cubical zigzag persistence computation (Reviewer rwpn). We have conducted and discussed sensitivity analysis of \epsilon-graph construction and learning (Reviewer rwpn). We have provided additional clarifications of the rasterization process (Reviewer rwpn).

6. We have added an explanation to clarify the motivation and problem analysis in this paper (Reviewer UFd9).

7. We have conducted additional experiments on 311 service data with (a) noisy data and (b) missing data and shown our TEN-DM achieves consistently promising performance on both spatial and temporal dimensions (Reviewer UFd9).

Best,

Authors

---

### Meta-Review · Area_Chair_Ls6H · 2026-01-06

**Summary:**

This paper proposes a Topology-ENhanced Diffusion Model (TEN-DM) for forecasting events in spatio-temporal point processes by integrating graph neural networks, zigzag persistence from topological data analysis, and a conditional diffusion model.

**Reviewer Concerns:**

The idea is considered new in integrating zigzag persistence with diffusion models for the first time in this domain (Wz4A), effective (rwpn). Although some weaknesses remain, including insufficient motivation and problem analysis for the core design choices (UFd9), exceptional model complexity raising practicality and scalability concerns (rwpn, Wz4A), and underspecified details in the critical data rasterization step (rwpn), the authors have addressed the main issues in their response and the clarity still need to be improved in the revision.

**Reviewer Scores:**

rwpn would keep the score unchanged.

Wz4A would raise the score based on the rebuttal.

UFd9 would not change score as the concerns are primarily on clarity.

---

### Decision · Program_Chairs · 2026-01-26

Accept (Poster)